# Deletion of a conserved Gata2 enhancer impairs haemogenic endothelium programming and adult Zebrafish haematopoiesis

Tomasz Dobrzycki[1], Christopher B. Mahony[2], Monika Krecsmarik[1,3], Cansu Koyunlar[4], Rossella Rispoli[1,5], Joke Peulen-Zink [4], Kirsten Gussinklo[4], Bakhta Fedlaoui[2], Emma de Pater [4], Roger Patient[1,3] & Rui Monteiro [1,2,3 ✉]

Gata2 is a key transcription factor required to generate Haematopoietic Stem and Progenitor Cells (HSPCs) from haemogenic endothelium (HE); misexpression of Gata2 leads to haematopoietic disorders. Here we deleted a conserved enhancer (i4 enhancer) driving pan-endothelial expression of the zebrafish *gata2a* and showed that Gata2a is required for HE programming by regulating expression of *runx1* and of the second Gata2 orthologue, *gata2b*. By 5 days, homozygous *gata2a*$^{\Delta i4/\Delta i4}$ larvae showed normal numbers of HSPCs, a recovery mediated by Notch signalling driving *gata2b* and *runx1* expression in HE. However, *gata2a*$^{\Delta i4/\Delta i4}$ adults showed oedema, susceptibility to infections and marrow hypo-cellularity, consistent with bone marrow failure found in GATA2 deficiency syndromes. Thus, *gata2a* expression driven by the i4 enhancer is required for correct HE programming in embryos and maintenance of steady-state haematopoietic stem cell output in the adult. These enhancer mutants will be useful in exploring further the pathophysiology of GATA2-related deficiencies in vivo.

---

[1] MRC Molecular Haematology Unit, MRC Weatherall Institute of Molecular Medicine, John Radcliffe Hospital, University of Oxford, Oxford OX3 9DS, UK. [2] Institute of Cancer and Genomic Sciences, College of Medical and Dental Sciences, University of Birmingham, Birmingham B15 2TT, UK. [3] BHF Centre of Research Excellence, Oxford, UK. [4] Department of Hematology, Erasmus MC, Rotterdam, The Netherlands. [5] Division of Genetics and Molecular Medicine, NIHR Biomedical Research Centre, Guy's and St Thomas' NHS Foundation Trust and King's College London, London, UK. ✉email: r.monteiro@bham.ac.uk

Haematopoietic stem cells (HSCs) are the source of all blood produced throughout the lifetime of an organism. They are capable of self-renewal and differentiation into progenitor cells that generate specialised blood cell types. DNA-binding transcription factors are fundamental players in the inception of the haematopoietic system as it develops in the embryo, but also play a crucial role in maintaining homeostasis of the haematopoietic system in the adult organism. They coordinate differentiation, proliferation and survival of haematopoietic cells and ensure their levels are appropriate at all times throughout life. Misexpression of key transcription factors may thus lead to a failure to produce HSCs or, alternatively, to haematopoietic disorders and eventually leukaemia. Therefore, understanding how transcription factors drive the haematopoietic process provides opportunities for intervention when haematopoiesis is dysregulated.

The development of blood occurs in distinct waves: primitive, pro-definitive and definitive, each of them characterised by the generation of blood progenitors in a specific location and restricted in time, where the definitive wave produces multi-lineage self-renewing HSCs[1]. The specification of HSCs initiates in cells with arterial characteristics[2] and proceeds through an endothelial intermediate, termed the haemogenic endothelium (HE)[3]. In zebrafish and other vertebrates, expression of *runx1* defines the *bona fide* HE population[4,5]. Haematopoietic stem and progenitor cells (HSPCs) emerge from the HE by endothelial-to-haematopoietic transition (EHT), both in zebrafish and in mice[6–8]. They arise between 28 and 48 h post fertilisation (hpf) from the HE in the ventral wall of the dorsal aorta (DA)[9], the analogue of the mammalian aorta-gonad-mesonephros (AGM)[10]. After EHT, the HSCs enter the bloodstream through the posterior cardinal vein (PCV)[9] to colonise the caudal haematopoietic tissue (CHT), the zebrafish equivalent of the mammalian foetal liver[11]. Afterwards the HSCs migrate again within the bloodstream to colonise the kidney marrow (WKM) and thymus[9], the final niche for HSCs, equivalent to the bone marrow in mammals[1].

Gata2 is a key haematopoietic transcription factor (TF) in development. In humans, *GATA2* haploinsufficiency leads to blood disorders, including MonoMAC syndrome (Mono-cytopenia, Mycobacterium avium complex) and myelodysplastic syndrome (MDS)[12,13]. While its presentation is variable, Mono-MAC syndrome patients always show cytopenias, ranging from mild to severe, and hypocellular bone marrow[13,14]. These patients are susceptible to mycobacterial and viral infections, and have a propensity to develop MDS and Acute Myeloid Leukaemia (AML), with a 75% prevalence and relatively early onset at age 20[13].

*Gata2* knockout mice are embryonic lethal and die by E10.5[15]. Conditional *Gata2* knockout under the control of the endothelial *VE-cad* promoter abolished the generation of intra-aortic clusters[16], suggesting that Gata2 is required for HSPC formation. Further studies in the mouse revealed a decrease in HSC numbers in *Gata2* heterozygous mutants, but also a dose-dependency of adult HSCs on Gata2[17].

*Gata2* expression in the endothelium is regulated by an intronic enhancer element termed the +9.5 enhancer[18,19]. Deletion of this enhancer results in the loss of HSPC emergence from HE, leading to lethality by E14[19]. The same element is also mutated in 10% of all the MonoMAC syndrome patients[12].

Because of a partial genome duplication during the evolution of teleost fish, numerous zebrafish genes exist in the form of two paralogues, including *gata2*[20]. This provides an opportunity to separately identify the temporally distinct contributions made by each Gata2 orthologue. *Gata2a* and *gata2b* are only 57% identical and are thought to have undergone evolutionary sub-functionalisation from the ancestral vertebrate *Gata2* gene[21,22].

*Gata2b* is expressed in HE from 18hpf and is thought to regulate *runx1* expression in HE[21]. Lineage tracing experiments showed that *gata2b*-expressing HE cells gave rise to HSCs in the adult[21]. Similar to the mouse Gata2, *gata2b* expression depends on Notch signalling and is a *bona fide* marker of HE, currently regarded as the functional 'haematopoietic homologue' of Gata2 in zebrafish[21]. By contrast, *gata2a* is expressed in all endothelial cells and in the developing central nervous system[21,23]. Homozygous *gata2a*[um27] mutants showed arteriovenous shunts in the dorsal aorta at 48hpf[24]. However, *gata2a* is expressed at 11hpf in the haemangioblast population in the posterior lateral mesoderm (PLM) that gives rise to the arterial endothelial cells in the trunk[25], well before *gata2b* is expressed in HE. This suggests that *gata2a* might play a role in endothelial and HE programming and thus help to elucidate an earlier role for Gata2 in HSC development.

Here we show that the *gata2a* locus contains a conserved enhancer in its 4th intron, corresponding to the described +9.5 enhancer in the mouse Gata2 locus[18,19]. Using CRISPR/Cas9 genome editing, we demonstrated that this region, termed the i4 enhancer, is required for endothelial-specific *gata2a* expression. Homozygous mutants (*gata2a*[Δi4/Δi4] mutants) showed decreased expression of the HE-specific genes *runx1* and *gata2b*. Thus, endothelial expression of *gata2a*, regulated by the i4 enhancer, is required for *gata2b* and *runx1* expression in the HE. Strikingly, their expression recovers and by 48hpf, the expression of haematopoietic markers in *gata2a*[Δi4/Δi4] mutants is indistinguishable from wild-type siblings. We have demonstrated that this recovery is mediated by an independent input from Notch signalling, sufficient to recover *gata2b* and *runx1* expression in HE and thus HSPC emergence by 48hpf. We conclude that *runx1* and *gata2b* are regulated by two different inputs, one Notch-independent input from Gata2a and a second from the Notch pathway, acting as a fail-safe mechanism for the initial specification of HSPCs in the absence of the input by Gata2a. Despite the early rescue, *gata2a*[Δi4/Δi4] adults showed increased susceptibility to infections, oedema, a hypocellular WKM and neutropenia, a phenotype resembling key features of GATA2 deficiency syndromes in humans. We conclude that Gata2a is required for HE programming in the embryo and to maintain the steady-state haematopoietic output from adult HSPCs and that this function requires the activity of the i4 enhancer.

## Results

**Analysis of open chromatin regions in the *gata2a* locus**. Because Gata2 genes are duplicated in zebrafish, we set out to unpick the different roles Gata2a and Gata2b play during HSC generation and homeostasis by identifying their regulatory regions. Analysis of sequence conservation revealed that one region within the fourth intron of the zebrafish *gata2a* locus was conserved in vertebrates, including mouse and human (Fig. 1a–c). This region, which we termed 'i4 enhancer', corresponds to the endothelial +9.5 Gata2 enhancer identified previously in the mouse[18,19] and human[26]. Notably, the *gata2b* locus did not show broad conservation in non-coding regions (Supplementary Fig. 1a).

To investigate whether the i4 element was a potentially active enhancer, we first performed ATAC-seq[27] to identify open chromatin regions in endothelial cells (ECs) in zebrafish. We used a Tg(*kdrl*:GFP) transgenic line that expresses GFP in all endothelium[28] and isolated the higher GFP-expressing ECs (*kdrl*:GFP[high], termed *kdrl*:GFP[+] for simplicity) as this fraction was enriched for endothelial markers compared to the *kdrl*:GFP[low] fraction (Supplementary Fig. 1b, c). Principal Component Analysis on the ATAC-seq data from 26hpf *kdrl*:GFP[+] cells

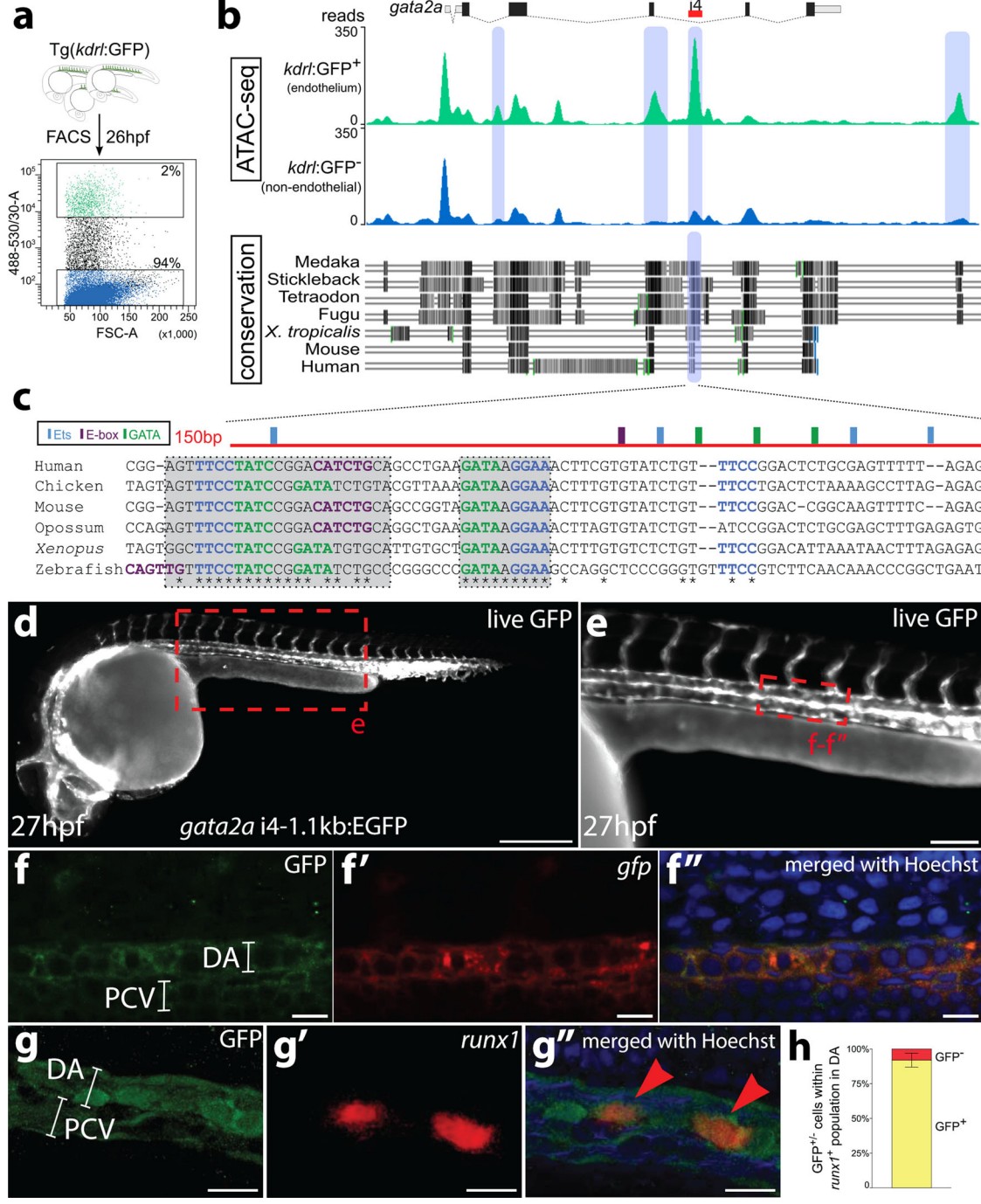

**Fig. 1 The i4 enhancer in the *gata2a* locus is conserved and drives pan-endothelial expression of a GFP reporter in zebrafish. a** *Kdrl*:GFP⁺ (green) and *kdrl*:GFP⁻ (blue) cells were FACS-sorted from 26hpf embryos and used for preparation of ATAC-seq libraries. **b** The image of the mapped reads represents stacked means of two biological ATAC-seq replicates. Differential peak analysis identified four chromatin regions (blue shading) in the locus of *gata2a* that are significantly more open in the *kdrl*:GFP⁺ population (*p* < 0.0001). A region in the fourth intron (termed i4 enhancer) is conserved throughout vertebrates. Black and grey shading denotes regions of high conservation between the species analysed. **c** The highly conserved 150 bp region (red) contains putative transcription factor binding sites, mapped computationally. Light blue: Ets binding sites; purple: E-box binding sites; green: GATA binding sites; asterisks: conserved residues. **d** Widefield fluorescent image of a live Tg(*gata2a*-i4-1.1 kb:GFP) zebrafish embryo at 27hpf showing GFP fluorescence in the endothelial cells and in the heart (endocardium). **e** Higher magnification image of the trunk of the embryo from panel **d**. **f–f″** Confocal images of a trunk fragment of a Tg(*gata2a*-i4-1.1 kb:GFP) embryo immunostained with anti-GFP antibody (**f**) and probed for *gfp* mRNA (**f′**) at 25hpf. **f″** Merged images from panels **f–f′** with Hoechst nuclear staining in blue, showing complete overlap of GFP protein and mRNA. **g–g″** Confocal images of the dorsal aorta (DA) and posterior cardinal vein (PCV) of a Tg(*gata2a*-i4-1.1 kb:GFP) embryo immunostained with anti-GFP antibody (**g**) and probed for *runx1* mRNA (**g′**) at 25hpf. See panel **e** for approximate position within the embryo. **g″** Merged images from panels **g–g′**, also showing Hoechst nuclear staining in blue. **h** Counting of the *runx1*⁺ cells represented in panels **g′–g″** in 25 embryos shows that >90% of *runx1*⁺ cells are also GFP⁺. *N* = 3. Error bars: ± SD. See also Supplementary Fig. 1.

($n = 2$) and $kdrl$:GFP$^-$ cells ($n = 4$) revealed strong differences between the open chromatin regions in the two cell populations, further supported by a correlation analysis (Supplementary Fig. 1d–f). 78,026 peaks were found in common between replicates of the ATACseq in $kdrl$:GFP$^+$ cells (Supplementary Fig. 1g). 44,025 peaks were differentially expressed between the $kdrl$:GFP$^+$ and $kdrl$:GFP$^-$ fractions (Supplementary Fig. 1h). An analysis of known motifs present in the $kdrl$:GFP$^+$ population revealed an enrichment for the ETS motif (Supplementary Fig. 1i). ETS factors are essential regulators of gene expression in endothelium[29]. In addition, we performed gene ontology (GO) term analysis on the peaks showing >3-fold enrichment or depletion in ECs (Supplementary Fig. 1j–l). As expected, non-ECs showed a broad range of GO terms whereas EC-enriched peaks were associated with terms like angiogenesis or blood vessel development (Supplementary Fig. 1k, l).

Differential peak analysis in the $gata2a$ locus identified four differentially open sites within a 20 kb genomic region (Fig. 1b), including one peak in intron 4 corresponding to the predicted i4 enhancer. It contained a core 150 bp-long element that included several binding motifs for the GATA, E-box and Ets transcription factor families (Fig. 1b). Although the positioning of the E-box site relative to the adjacent GATA site differs in zebrafish and mammals (Fig. 1b, c), the necessary spacer distance of ~9 bp between the two sites[30] was conserved. Thus, this site may be a target for TF complexes containing an E-box-binding factor and a GATA family TF.

Thus, the intronic enhancer (i4) identified in the zebrafish $gata2a$ locus is accessible to transposase in endothelial cells and contains highly conserved binding sites for key haematopoietic transcription factors, suggesting that genetic regulation of $gata2a$ expression in zebrafish HE is a conserved feature of vertebrate $gata2$ genes.

**The $gata2a$-i4 enhancer drives GFP expression in endothelium**. To investigate the activity of the $gata2a$-i4 enhancer in vivo, the conserved genomic 150 bp region (Fig. 1b, c), together with flanking ±500 bp ($gata2a$-i4-1.1 kb:GFP) or ±150 bp ($gata2a$-i4-450 bp:GFP) was cloned into a Tol2-based reporter E1b:GFP construct[31] and used to generate stable transgenic lines (Supplementary Fig. 2). The earliest activity of the enhancer was observed at the 14-somite stage (14ss), when $gfp$ mRNA was detected in the PLM (Supplementary Fig. 2a, b). After 22hpf, the reporter signal was pan-endothelial (Fig. 1d–e, Supplementary Fig. 2c–i). Around 27hpf, higher intensities of GFP fluorescence and correspondingly higher levels of $gfp$ mRNA were visible in the floor of the DA (Fig. 1d–e, Supplementary Fig. 2e–h). While the GFP protein was still visible in the vasculature around 3dpf, it was likely carried over from earlier stages, since the $gfp$ mRNA was not detectable any more (Supplementary Fig. 2i, j). We focussed our subsequent analysis on the $gata2a$-i4-1.1 kb:GFP transgenics as they showed stronger expression of the transgene. At 25hpf, the expression of GFP protein and $gfp$ mRNA overlapped completely in the endothelial cells of the DA (Fig. 1f–f″). Overall, these data confirm that the i4 enhancer is active in vivo in endothelial cells at the correct time to regulate definitive haematopoiesis. The endothelial activity of the corresponding +9.5 enhancer was also observed in mouse embryos[18], indicating functional conservation of the $gata2a$-i4 enhancer across vertebrates.

To further characterise the enhancer activity in vivo, Tg ($gata2a$-i4-1.1 kb:GFP) embryos were stained for $gata2a$ mRNA and for GFP protein (Supplementary Fig. 2k–o). We found a large overlap between $gata2a^+$ and GFP$^+$ cells at 30hpf in the DA, with a small proportion of GFP$^+$ cells that did not express $gata2a$

mRNA (<5%, Supplementary Fig. 2o). This could suggest that some cells require activity of other endothelial enhancers to trigger transcription of $gata2a$ or that $gfp$ mRNA has a longer half-life than $gata2a$ mRNA. Importantly, the GFP signal was absent in $gata2a$-expressing neural cells (Supplementary Fig. 2l–n), indicating that the i4 enhancer is specifically active in (haemogenic) endothelial cells.

Next we examined the expression of the HE marker $runx1$[4] in $gata2a$-i4-1.1 kb:GFP embryos at 25hpf. At this stage, over 90% of $runx1^+$ cells were GFP$^+$ (Fig. 1g–h). We conclude that the GFP expression under the $gata2a$-i4 enhancer marks the majority of the HE population.

**Endothelial $gata2a$ required $gata2a$-i4 enhancer activity**. To investigate whether endothelial-specific expression of $gata2a$ is required for definitive haematopoiesis, we deleted the conserved $gata2a$-i4 enhancer using CRISPR/Cas9 genome editing[32]. We generated a deletion mutant lacking 231 bp of the i4 enhancer (Supplementary Fig. 3a–c) and named it $gata2a^{\Delta i4/\Delta i4}$. Homozygous $gata2a^{\Delta i4/\Delta i4}$ mutants showed decreased levels of $gata2a$ expression in endothelial cells when compared to wild-type embryos (Fig. 2a, b). By contrast, $gata2a$ expression in the neural tube appeared unaffected in the $gata2a^{\Delta i4/\Delta i4}$ mutants (Fig. 2a, b). At 28hpf, expression of the pan-endothelial marker $kdrl$ was indistinguishable between wild-type and $gata2a^{\Delta i4/\Delta i4}$ mutants (Fig. 2c, d). To verify these results, we crossed homozygous $gata2a^{\Delta i4/\Delta i4}$ mutants to Tg($kdrl$:GFP) transgenics and analysed vascular morphology. $Gata2a^{\Delta i4/\Delta i4}$ embryos showed no gross vascular abnormalities at 48hpf as assessed by the expression of the Tg($kdrl$:GFP) transgene (Fig. 2e, f).

Next, we isolated endothelial cells from Tg($kdrl$:GFP) and Tg($kdrl$:GFP); $gata2a^{\Delta i4/\Delta i4}$ embryos by FACS (Fig. 2g) at 23hpf and 30hpf and confirmed by qRT-PCR that the endothelial markers $kdrl$, $dld$ and $dll4$ were unaffected in $gata2a^{\Delta i4/\Delta i4}$ embryos (Supplementary Fig. 3d–f). The arterial marker $efnb2a$[33] was decreased at 23hpf in $gata2a^{\Delta i4/\Delta i4}$ mutants but recovered by 30hpf (Supplementary Fig. 3g). In addition, $gata2a$ was significantly decreased in the $kdrl$:GFP$^+$ ECs in 23hpf $gata2a^{\Delta i4/\Delta i4}$ embryos compared to wild types (Fig. 2h). At 30hpf this decrease was not statistically significant. This was likely due to a decrease in expression of $gata2a$ that appears to occur in wild-type ECs during development, whereas $gata2a$ expression in mutants remained low (Fig. 2h). Importantly, there was no difference in $gata2a$ expression in the non-endothelial population ($kdrl$:GFP$^-$ cells) between wild-type and $gata2a^{\Delta i4/\Delta i4}$ mutants at either 23 hpf or 30hpf (Fig. 2i). Altogether, these data suggest that genomic deletion of the $gata2a$-i4 enhancer is sufficient to reduce expression of $gata2a$ specifically in endothelium.

**Gata2a regulates $runx1$ and $gata2b$ in haemogenic endothelium**. To investigate a potential role of $gata2a$ in HSC development, we compared the expression of $runx1$, the key marker of HE in zebrafish[4], in wild-type and $gata2a^{\Delta i4/\Delta i4}$ embryos. Quantitative in situ hybridization (ISH) analysis[34] showed that $runx1$ expression was decreased in $gata2a^{\Delta i4/\Delta i4}$ embryos at 24hpf (Supplementary Fig. 4a–c) and 28hpf compared to wild-type siblings (Fig. 3a–c). Further analysis in $kdrl$:GFP$^+$ ECs showed that this decrease in $runx1$ expression was already detectable at 23hpf in $gata2a^{\Delta i4/\Delta i4}$ mutants (Fig. 3d), at the onset of its expression in HE[35]. Thus, deletion of the $gata2a$-i4 enhancer results in impaired $runx1$ expression in the early stages of HE programming. This correlates well with decreased $runx1$ expression levels in +9.5$^{-/-}$ mouse AGM explants[19], further supporting the critical evolutionary role of the intronic enhancer of $Gata2$ in HSC specification.

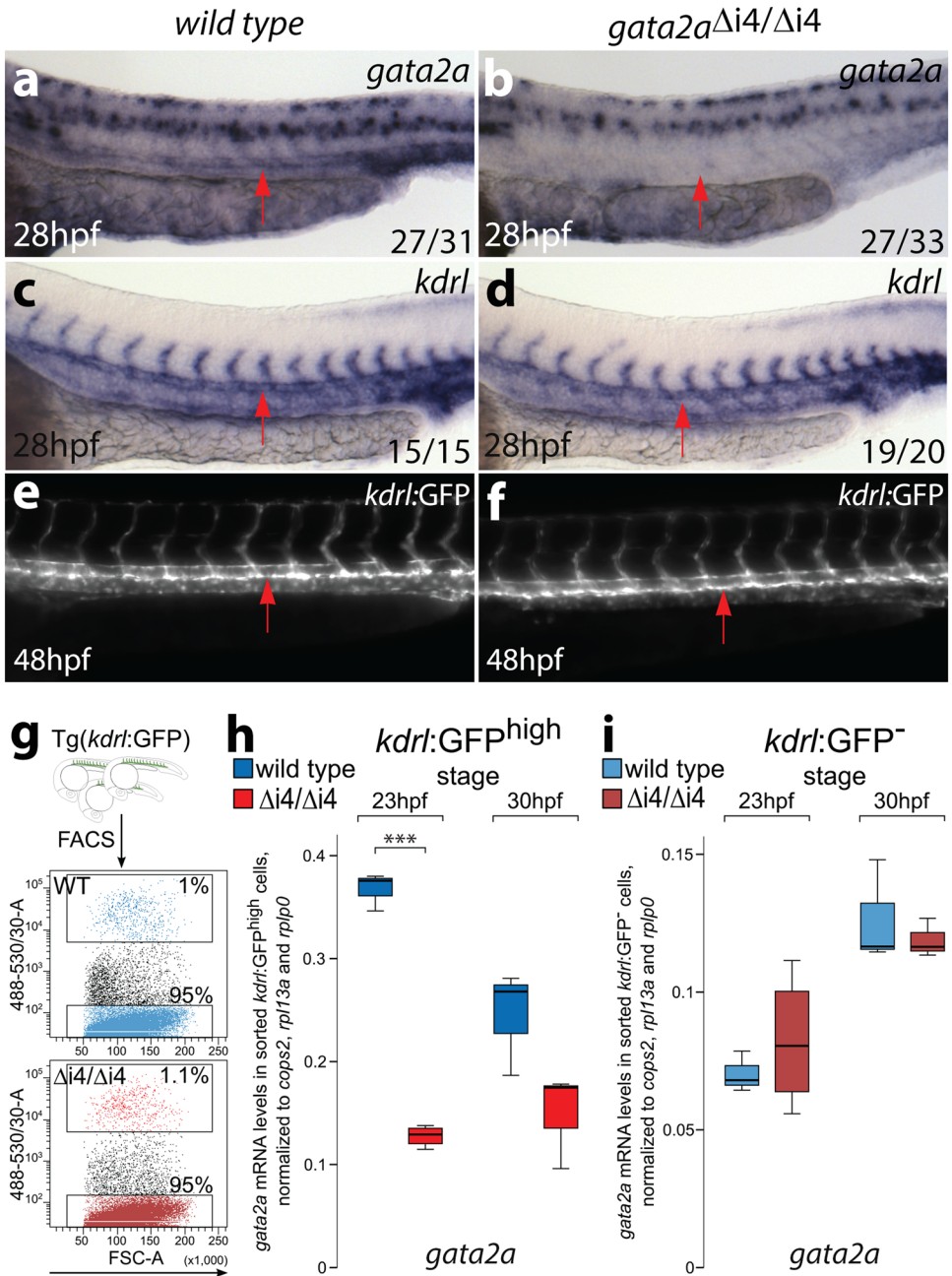

**Fig. 2 Deletion of the i4 enhancer in *gata2a*$^{\Delta i4/\Delta i4}$ mutants leads to reduced levels of *gata2a* mRNA in the endothelium. a, b** A significant majority of *gata2a*$^{\Delta i4/\Delta i4}$ mutants have reduced levels of *gata2a* mRNA in the dorsal aorta (arrows) at 28hpf, compared to wild-type siblings, as detected with in situ hybridization. ($X^2 = 10.720$, d.f. = 1, $p < 0.01$). The expression in the neural tube appears unaffected. **c, d** In situ hybridization for the endothelial marker *kdrl* at 28hpf reveals no difference between *gata2a*$^{\Delta i4/\Delta i4}$ mutants and wild-type siblings. The dorsal aorta (arrows) appears unaffected. **e, f** Live images of the trunks of 48hpf Tg(*kdrl*:GFP) and Tg(*kdrl*:GFP); *gata2a*$^{\Delta i4/\Delta i4}$ embryos show normal vascular morphology in the mutants. The endothelium of the dorsal aorta (arrows) appears normal in the *gata2a*$^{\Delta i4/\Delta i4}$ embryos. **g** *Kdrl*:GFP$^{high}$ and *kdrl*:GFP$^-$ cells were sorted from non-mutant (WT, blue) and *gata2a*$^{\Delta i4/\Delta i4}$ (red) embryos carrying the Tg(*kdrl*:GFP) transgene. **h, i** qRT-PCR on RNA isolated from the sorted *kdrl*:GFP$^{high}$ or *kdrl*:GFP- cells (panel **g**) shows decreased levels of *gata2a* mRNA in the endothelium of *gata2a*$^{\Delta i4/\Delta i4}$ mutants at 23hpf ($t = 20.026$, d.f. = 5, $p < 0.001$) compared to wild-type. At 30hpf this difference is not statistically significant ($t = 2.146$, d.f. = 4, $p = 0.098$). There is no difference in *gata2a* mRNA levels in non-endothelial cells between wild-type and *gata2a*$^{\Delta i4/\Delta i4}$ mutants (23hpf: t = 0.69, d.f. = 5, $p > 0.5$; 30hpf: t = 0.618, d.f. = 4, $p > 0.5$). N = 4 for *gata2a*$^{\Delta i4/\Delta i4}$ at 23hpf, N = 3 for other samples. Note different scales of expression levels. ***$p < 0.001$. See also Supplementary Fig. 2.

Next, we tested whether Gata2a could act upstream of *gata2b* by measuring its expression in *gata2a*$^{\Delta i4/\Delta i4}$ embryos. Quantitation of the ISH signal showed that *gata2b* expression was decreased in *gata2a*$^{\Delta i4/\Delta i4}$ embryos compared to wild-type siblings at 26hpf (Supplementary Fig. 4d) and 28hpf (Fig. 3e–g),

but recovered to wild-type levels by 30hpf (Supplementary Fig. 4e). Accordingly, *kdrl*:GFP$^+$; *gata2a*$^{\Delta i4/\Delta i4}$ cells express significantly lower levels of *gata2b* mRNA than the wild-type *kdrl*:GFP$^+$ endothelial population at 23hpf, but not at 30hpf (Fig. 3h). These data suggest that endothelial expression of *gata2a*

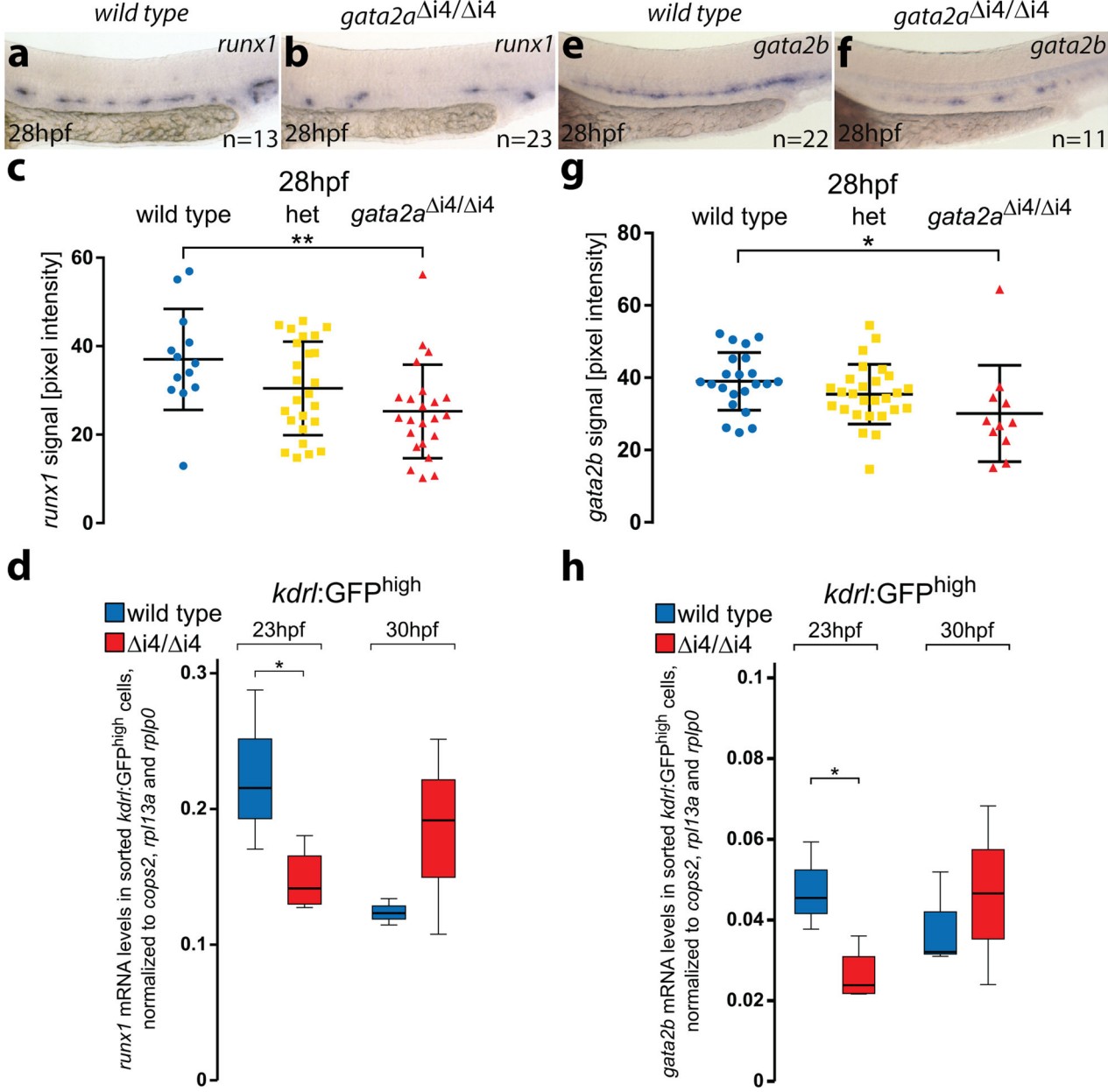

**Fig. 3 Loss of *gata2a* expression in the endothelium of *gata2a*$^{\Delta i4/\Delta i4}$ mutants leads to decreased levels of *runx1* and *gata2b* in the HE. a, b** In situ hybridization for *runx1* expression in the HE of wild-type and *gata2a*$^{\Delta i4/\Delta i4}$ embryos at 28hpf. **c** Quantification of the *runx1* in situ hybridization signal from wild-type (blue), heterozygous *gata2a*$^{+/\Delta i4}$ (het, yellow) and *gata2a*$^{\Delta i4/\Delta i4}$ (red) siblings at 28hpf shows significant decrease in *runx1* pixel intensity in the DA in the homozygous mutants compared to wild-type ($\mu_{wt} = 34.8$, $\mu_{mut} = 25.3$; $F = 4.956$, d.f. $= 2$, 58; ANOVA),**$p < 0.01$. n = 14, wild-type; $n = 25$, het; $n = 23$, *gata2a*$^{\Delta i4/\Delta i4}$. Error bars: mean ± SD. **d** qRT-PCR on RNA isolated from the sorted *kdrl*:GFP$^+$ cells shows decreased levels of *runx1* mRNA in the endothelium of *gata2a*$^{\Delta i4/\Delta i4}$ mutants at 23hpf ($t = 2.585$, d.f. $= 5$, $p < 0.05$) but not at 30hpf ($t = 1.326$, d.f. $= 4$, $p > 0.2$), compared to wild-type. $N = 4$ for *gata2a*$^{\Delta i4/\Delta i4}$ at 23hpf, $N = 3$ for other samples. Note different scales of expression levels. *$p < 0.05$. **e, f** *Gata2b* expression in the HE of wild-type and *gata2a*$^{\Delta i4/\Delta i4}$ embryos at 28hpf. **g** Quantification of the *gata2b* mRNA signal, detected by in situ hybridization, from wild-type (blue), heterozygous *gata2a*$^{+/\Delta i4}$ (het; yellow) and *gata2a*$^{\Delta i4/\Delta i4}$ (red) siblings at 28hpf shows significant decrease in *gata2b* pixel intensity in the DA in the homozygous mutants compared to wild-type ($\mu_{wt} = 39$, $\mu_{mut} = 30.1$; $F = 5.05$, d.f. $= 2$, 54; ANOVA), *$p < 0.05$. n = 22, wild-type; $n = 24$, het; $n = 11$, *gata2a*$^{\Delta i4/\Delta i4}$. Error bars: mean ± SD. **h** qRT-PCR in sorted *kdrl*:GFP$^+$ cells showed decreased levels of *gata2b* mRNA in the endothelium of *gata2a*$^{\Delta i4/\Delta i4}$ mutants at 23hpf ($t = 3.334$, d.f. $= 5$, $p < 0.05$) but not at 30hpf ($t = 0.373$, d.f. $= 4$, $p > 0.7$), compared to wild-type. $N = 4$ for *gata2a*$^{\Delta i4/\Delta i4}$ at 23hpf, $N = 3$ for other samples. *$p < 0.05$. See also Supplementary Figs. 3 and 4.

is required upstream of *gata2b* and *runx1* for the proper specification of HE, uncovering a previously unrecognized role for Gata2a in definitive haematopoiesis.

**Recovery of embryonic HSC activity in *gata2a*$^{\Delta i4/\Delta i4}$ mutants.** The qPCR analysis in sorted *kdrl*:GFP$^+$; *gata2a*$^{\Delta i4/\Delta i4}$ and wild-type *kdrl*:GFP$^+$ ECs (Fig. 3d) already suggested a recovery of *runx1* expression from 30hpf. Of note, the *kdrl*:GFP$^+$ population likely includes the *kdrl*$^+$, *runx1*-expressing erythromyeloid progenitors (EMPs) located in the caudal region[36]. This region was not included in the quantification of ISH but cannot be separated by sorting for *kdrl*:GFP$^+$ and could thus explain the discrepancy between image

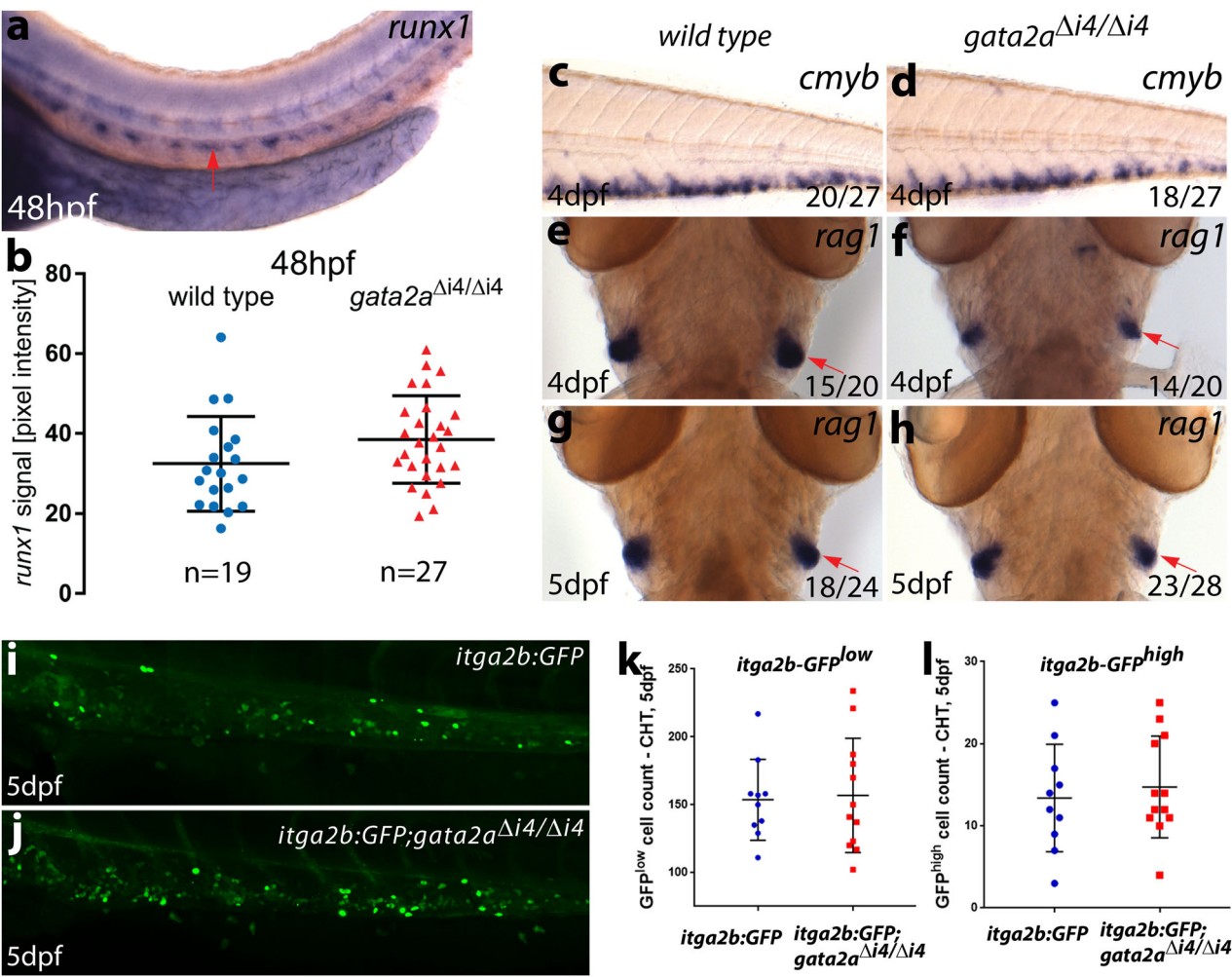

**Fig. 4 Gata2a$^{\Delta i4/\Delta i4}$ mutants display a recovery of the initial haematopoietic defects from 48hpf. a** Representative image of *runx1* expression in the trunk of a wild-type embryo at 48hpf showing *runx1* mRNA in the dorsal aorta (arrow). **b** Quantification of the *runx1* in situ hybridization signal in wild-type (blue) and *gata2a$^{\Delta i4/\Delta i4}$* mutants (red) siblings at 48hpf. There is no significant difference in *runx1* pixel intensity in the DA between the homozygous mutants and wild-type ($\mu_{wt}$ = 33.1, $\mu_{mut}$ = 37.5, $t$ = 1.410, d.f. = 44, $p$ = 0.17. $n$ = 19, wild-type; $n$ = 27, *gata2a$^{\Delta i4/\Delta i4}$*). Error bars: mean ± SD. **c, d** In situ hybridization for *cmyb* in the CHT. We detected no difference in expression between wild-type and *gata2a$^{\Delta i4/\Delta i4}$* siblings at 4dpf. (**e–h**) In situ hybridization (ventral view) for *rag1* in the thymii, showing a slight decrease (relative to wild-type) in *rag1* (red arrows) in approximately half of the homozygous mutant embryos at 4dpf. This effect is absent at 5dpf. (**i, j**) Maximum projections of *itga2b*:GFP transgenic embryos in the CHT at 5dpf in **i** wild-type and **j** *gata2a$^{\Delta i4/\Delta i4}$* siblings. **k** HSPC (*itga2b*:GFP$^{low}$) counts in the CHT of wild-type ($n$ = 10) and *gata2a$^{\Delta i4/\Delta i4}$* mutants ($n$ = 12) at 5dpf. No difference was detected between genotypes ($\mu_{wt}$ = 153.5; $\mu_{mut}$ = 145.5; $p$ = 0.98, Mann-Whitney test). **l** Thrombocyte (*itga2b*:GFP$^{high}$) counts in the CHT of wild-type ($n$ = 10) and *gata2a$^{\Delta i4/\Delta i4}$* mutants ($n$ = 12) at 5dpf. No difference was detected between genotypes ($\mu_{wt}$ = 13; $\mu_{mut}$ = 13; $p$ = 0.71, Mann-Whitney test). Error bars: median ± SD.

quantification and qRT-PCR. To further characterize the haemato-poietic phenotype in the *gata2a$^{\Delta i4/\Delta i4}$* mutants, we tested whether expression of markers of haematopoietic activity in the embryo was affected from 48hpf onwards (Fig. 4).

At 48hpf, the expression of *runx1* in the DA showed no significant difference between *gata2a$^{\Delta i4/\Delta i4}$* mutants and wild-type controls (Fig. 4a, b). These data suggest that the decrease of *runx1* expression at early stages of HE programming in *gata2a$^{\Delta i4/\Delta i4}$* mutants is transient and recovers by 2dpf. Indeed, analysis of the HSPC marker *cmyb*[11] in the CHT at 4dpf showed no differences between *gata2a$^{\Delta i4/\Delta i4}$* and wild-type larvae (Fig. 4c, d). Expression of the T-cell progenitor marker *rag1* in the thymus[37] showed that around half of the *gata2a$^{\Delta i4/\Delta i4}$* larvae had reduced *rag1* expression at 4dpf compared to wild-type (Fig. 4e, f). This effect was absent at 5dpf (Fig. 4g, h), suggesting that HSPC activity was normal in *gata2a$^{\Delta i4/\Delta i4}$* mutants from 4dpf onwards. Next, we crossed the *gata2a$^{\Delta i4/\Delta i4}$* mutants to Tg

(*itga2b*:GFP) transgenics, where *itga2b*-GFP$^{high}$ and *itga2b*-GFP$^{low}$ cells in the CHT mark thrombocytes and HSPCs, respectively[9,38]. Our analysis revealed no difference in *itga2b*-GFP$^{low}$ HSPC or *itga2b*-GFP$^{high}$ thrombocyte numbers in the CHT region at 5dpf between wild-type and *gata2a$^{\Delta i4/\Delta i4}$* mutants (Fig.4i–l). Taken together, our data suggest that endothelial *gata2a* expression mediated by the i4 enhancer is required for the initial expression of *gata2b* and *runx1* in the HE but largely dispensable after 2dpf.

**Notch recovers haematopoiesis in gata2a$^{\Delta i4/\Delta i4}$ mutants.** The recovery of *gata2b* expression by 30hpf (Fig. 3h, Supplementary Fig. 4e) coincides temporally with the observed decrease in *gata2a* in wild-type endothelial cells (Fig. 2h). Thus, we reasoned that other regulators of *gata2b* might compensate for the lack of endothelial *gata2a* in *gata2a$^{\Delta i4/\Delta i4}$* mutants and thus lead to a recovery of the initial haematopoietic phenotype. Therefore, we

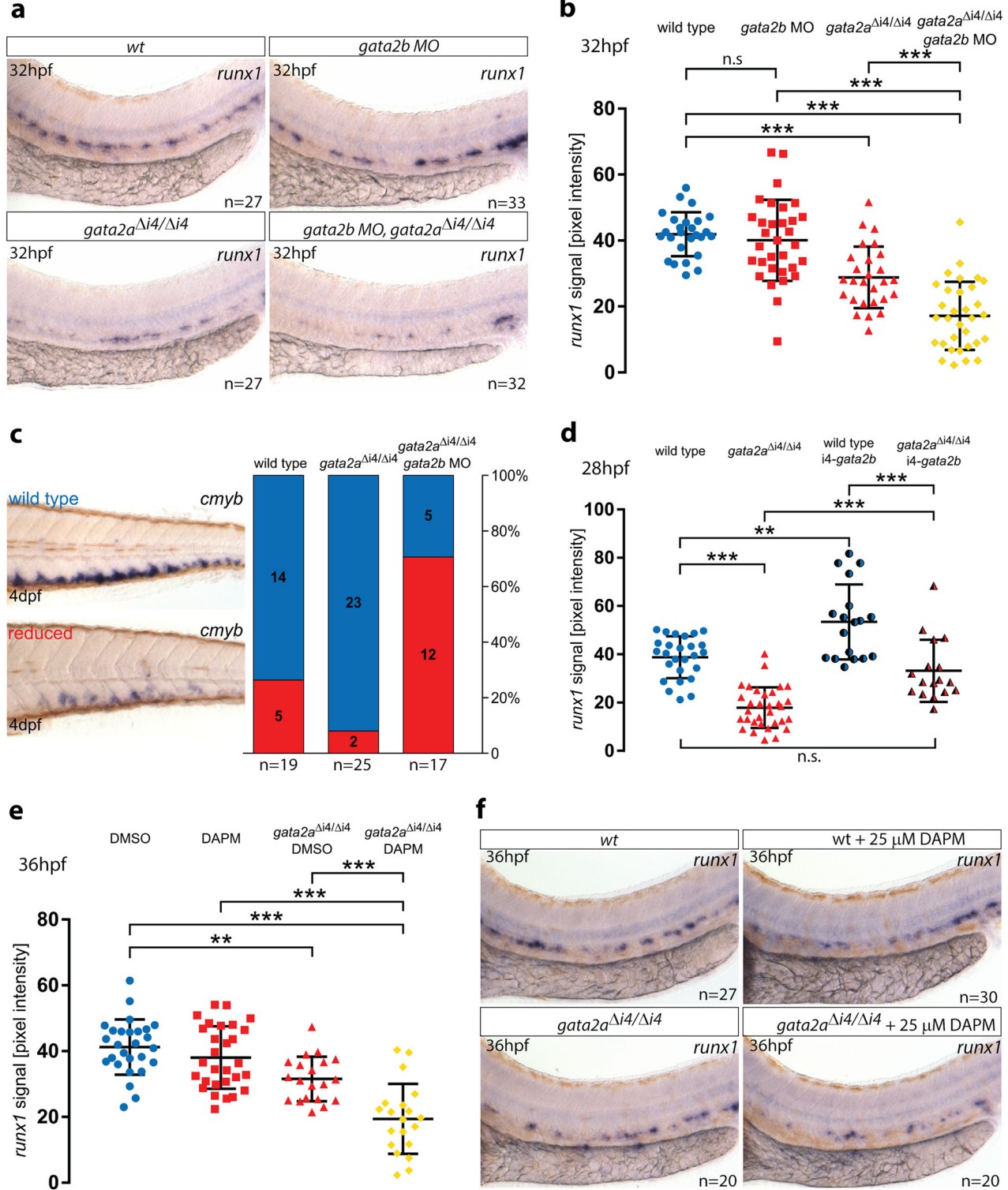

investigated whether the loss of *gata2b* in *gata2a*$^{\Delta i4/\Delta i4}$ background resulted in a more severe haematopoietic phenotype than observed in the *gata2a*$^{\Delta i4/\Delta i4}$ mutants. For this, we injected *gata2a*$^{\Delta i4/\Delta i4}$ and wild-type controls with a suboptimal amount (7.5 ng) of a *gata2b* morpholino oligonucleotide (MO)[21]. Quantitative ISH analysis confirmed that this amount of *gata2b* MO had no effect on *runx1* expression at 32hpf (Fig. 5a, b). As expected, *runx1* expression in *gata2a*$^{\Delta i4/\Delta i4}$ embryos was significantly reduced compared to wild-type siblings (Fig. 5a, b). *Gata2b* knockdown in *gata2a*$^{\Delta i4/\Delta i4}$ embryos further reduced

*runx1* expression (Fig. 5a, b). To test whether this stronger reduction of *runx1* at 32hpf affected later stages of embryonic haematopoiesis, we assessed *cmyb* expression in the CHT at 4dpf (Fig. 5c). We scored *cmyb* expression levels as 'wild-type' or 'reduced' and found that the 'reduced' embryos were substantially overrepresented in the *gata2a*$^{\Delta i4/\Delta i4}$ mutants injected with the *gata2b* MO, compared to wild-type fish and non-injected *gata2a*$^{\Delta i4/\Delta i4}$ siblings (Fig. 5c).

To verify whether Gata2b is required for definitive haematopoiesis downstream of Gata2a, we generated a frameshift truncating

**Fig. 5 Gata2b and Notch signalling are sufficient to recover haematopoietic markers in gata2a$^{\Delta i4/\Delta i4}$ mutants. a** Expression of *runx1* in HE at 32hpf in wild-type (wt), *gata2b* MO-injected (7.5 ng) wt embryos, *gata2a*$^{\Delta i4/\Delta i4}$ mutants and *gata2b* MO-injected (7.5 ng) *gata2a*$^{\Delta i4/\Delta i4}$ mutants. **b** Quantification of the *runx1* in situ hybridization (ISH) signal in wt, *gata2b* morphants, *gata2a*$^{\Delta i4/\Delta i4}$ mutants and *gata2a*$^{\Delta i4/\Delta i4}$ mutants injected with *gata2b* MO. *runx1* expression is decreased in *gata2a*$^{\Delta i4/\Delta i4}$ mutants ($\mu_{wt} = 41.9$, $\mu_{mut} = 28.9$; $F = 44.641$, d.f. = 3, 62.3; $p < 0.001$). *Gata2b* MO knockdown significantly decreases *runx1* in the DA of *gata2a*$^{\Delta i4/\Delta i4}$mutants ($\mu_{mut} = 28.9$, $\mu_{mut+MO} = 17.2$), but not wt embryos at 32hpf ($\mu_{wt} = 41.9$, $\mu_{MO} = 40.1$; $p = 0.89$, $p = 0.89$, Games-Howell post-hoc test, Welch's ANOVA). n = 27, *n* = 27, *gata2a*$^{\Delta i4/\Delta i4}$; *n* = 33, wt + *gata2b* MO; *n* = 32, *gata2a*$^{\Delta i4/\Delta i4}$ + *gata2b* MO. **c** Scoring *cmyb* expression at 4dpf in wt, *gata2a*$^{\Delta i4/\Delta i4}$ mutants and *gata2a*$^{\Delta i4/\Delta i4}$ mutant embryos injected with *gata2b* MO as wt (blue) or reduced (red). *Gata2b* MO knockdown (7.5 ng) inhibits the haematopoietic recovery of *gata2a*$^{\Delta i4/\Delta i4}$ mutants. ($X^2 = 18.784$, d.f. = 2, $p < 0.001$). **d** Quantification of the *runx1* ISH signal, from 28hpf wt embryos (blue), *gata2a*$^{\Delta i4/\Delta i4}$ mutants (red) and their siblings injected with a *gata2a*-i4-450bp:*gata2b* construct (shaded blue and red). Ectopic expression of *gata2b* increases *runx1* expression in the HE of wt embryos ($\mu_{wt} = 38.8$, $\mu_{wt+gata2b} = 53.4$; $p < 0.01$) and rescues *runx1* expression in the DA of *gata2a*$^{\Delta i4/\Delta i4}$ mutants to wt levels ($\mu_{mut} = 17.9$, $\mu_{mut+gata2b} = 33.2$; $p < 0.001$; $\mu_{wt} = 38.8$, $\mu_{mut+gata2b} = 33.2$; $p = 0.31$, Tukey HSD post-hoc test). n = 25, wt; *n* = 33, *gata2a*$^{\Delta i4/\Delta i4}$;n = 18, wt + *gata2a*-i4-450bp:*gata2b* construct; *n* = 17, *gata2a*$^{\Delta i4/\Delta i4}$ + *gata2a*-i4-450bp:*gata2b* construct. **e** Quantification of the *runx1* ISH signal at 36hpf in embryos treated with a suboptimal dose (25 μM) of the Notch inhibitor DAPM. 25 μM DAPM showed no effect on *runx1* expression in wt compared to DMSO-treated embryos ($\mu_{DMSO} = 40.5$, $\mu_{DAPM} = 38$; $p = 0.735$, Tukey HSD post-hoc test). DMSO-treated *gata2a*$^{\Delta i4/\Delta i4}$ mutants show a decrease in *runx1* expression ($\mu_{DMSO} = 40.5$, $\mu_{mut+DMSO} = 31.5$; $F = 25.774$, d.f. = 3, 91; ANOVA). DAPM treatment significantly reduced *runx1* expression in the DA *gata2a*$^{\Delta i4/\Delta i4}$ mutants ($\mu_{mut+DMSO} = 31.5$, $\mu_{mut+DAPM} = 19.4$). n = 27, wt +DMSO; *n* = 20, *gata2a*$^{\Delta i4/\Delta i4}$ + DMSO; *n* = 30, wt + DAPM; *n* = 20, *gata2a*$^{\Delta i4/\Delta i4}$ + DAPM. **f** Representative images of the average *runx1* expression at 36hpf in wt and *gata2a*$^{\Delta i4/\Delta i4}$ mutants treated with 25 μM DAPM. Error bars: mean ± SD. \*\*$p < 0.01$; \*\*\*$p < 0.001$. See also Supplementary Fig. 5.

mutant for Gata2b and incrossed *gata2a*$^{\Delta i4/+}$; *gata2b*$^{+/−}$ adults to investigate *cmyb* expression at 33hpf in their progeny. *Gata2b*$^{−/−}$ mutants showed a more severe decrease in *cmyb* expression than *gata2a*$^{\Delta i4/\Delta i4}$mutants (Supplementary Fig. 4f–i). Double *gata2b*$^{−/−}$; *gata2a*$^{\Delta i4/\Delta i4}$ mutants showed no further reduction in *cmyb* expression compared to *gata2b*$^{−/−}$ mutants, suggesting that Gata2a was not sufficient to drive *cmyb* expression in HE in the absence of Gata2b (Supplementary Fig. 4f–i). Taken together, we conclude that Gata2b is regulated by Gata2a and is required for definitive haematopoiesis.

Next, we tested whether forced ectopic expression of *gata2b* was sufficient to speed up the haematopoietic recovery of *gata2a*$^{\Delta i4/\Delta i4}$ embryos. Thus, we overexpressed *gata2b* under the control of the *gata2a*-i4-450bp enhancer in wild-type and *gata2a*$^{\Delta i4/\Delta i4}$ mutant embryos and measured *runx1* expression at 28hpf in the DA. *Gata2a*$^{\Delta i4/\Delta i4}$ embryos showed a significant decrease in *runx1* expression in comparison to wild-type (Fig. 5d). Ectopic expression of *gata2b* under the *gata2a*-i4 enhancer significantly increased *runx1* expression in wild-type and mutants (Fig. 5d). Importantly, it was sufficient to bring the *runx1* expression levels in the mutants up to the levels detected in uninjected wild-type embryos (Fig. 5d), demonstrating that *gata2b* alone was sufficient to drive *runx1* expression and drive the haematopoietic recovery in *gata2a*$^{\Delta i4/\Delta i4}$ mutants. Thus, *gata2b* can recover the definitive haematopoietic programme in the absence of endothelial *gata2a*.

Because the expression of *gata2b* is regulated by Notch signalling[21], we investigated whether inhibition of Notch would also prevent the haematopoietic recovery of *gata2a*$^{\Delta i4/\Delta i4}$ embryos. For this, we used the Notch inhibitor DAPM[39], and titrated it down to a suboptimal dose (25 μM) that did not significantly affect *runx1* expression at 30hpf in wild-type embryos (Supplementary Fig. 5a). This dose induced a small but measurable decrease in *gata2b* expression in DAPM-treated embryos while higher doses had a more robust effect (Supplementary Fig. 5b). Next, we treated wild-type and *gata2a*$^{\Delta i4/\Delta i4}$ mutant embryos with DAPM and measured *runx1* expression in the DA at 36hpf (Fig. 5e, f). Suboptimal DAPM treatment did not affect *runx1* expression in wild-type embryos (Fig. 5e, f), but *gata2a*$^{\Delta i4/\Delta i4}$ mutants showed lower *runx1* levels and DAPM treatment further reduced *runx1* expression (Fig. 5e, f). Treatment with 25 μM DAPM did not significantly affect *gata2b* expression at 36hpf in either wild-type or *gata2a*$^{\Delta i4/\Delta i4}$ mutant embryos (Supplementary Fig. 5c). These experiments suggested

that Notch signalling alone may be sufficient to rescue expression of HE markers in *gata2a*$^{\Delta i4/\Delta i4}$ mutants. Indeed, ectopic activation of Notch signalling in endothelium using a *fli1a*-NICD:GFP construct[40] led to increased *runx1* and *gata2b* expression in wild-type embryos (Supplementary Fig. 5d, e). When overexpressed in *gata2a*$^{\Delta i4/\Delta i4}$ mutants, *fli1a*-NICD:GFP rescued *runx1* expression to near wild-type levels at 26hpf, whereas *gata2b* was increased beyond normal levels independently of the genotype (Supplementary Fig. 5f, g). Taken together, these experiments confirm that Notch activity regulates *runx1* and *gata2b* in HE and is sufficient to drive haematopoietic recovery in *gata2a*$^{\Delta i4/\Delta i4}$ mutants. Thus, we conclude that HE programming requires two independent inputs on *runx1* and *gata2b* expression; one from Gata2a, driven in ECs by the i4 enhancer, and the other from Notch signalling, necessary and sufficient to drive HE programming even in the absence of *gata2a*.

**Impaired haematopoiesis in adult *gata2a*$^{\Delta i4/\Delta i4}$ mutants.** To investigate whether Gata2a plays a role in adult haematopoiesis, we first asked whether the *gata2a*-i4-1.1 kb:GFP reporter was active in haematopoietic cells in the adult. Whole kidney marrow (WKM) cells isolated from the transgenic fish showed that the i4 enhancer is active in haematopoietic cells previously defined by flow cytometry[41] as progenitors, lymphoid + HSPC (containing the HSPCs) and myeloid cells (Supplementary Fig. 6a–c). Accordingly, single cell transcriptional profiling showed higher levels of *gata2a* in HSPCs, progenitors, neutrophils and thrombocytes (Supplementary Fig. 6d–f)[42,43]. Consistent with this notion, we observed a high incidence of infections and heart oedemas in *gata2a*$^{\Delta i4/\Delta i4}$ adult fish, with over 25% suffering from one of these defects by 6 months of age, compared to <1% of wild-type fish (Fig. 6a–c). The heart oedemas and the infections are suggestive of lymphatic defects and immune deficiency as observed in human patients bearing genetic GATA2 haploinsufficiency syndromes such as MonoMAC syndrome[13]. Notably, around 10% of MonoMAC syndrome patients show mutations in the homologous enhancer region of *GATA2*[12,14].

Next, we counted the total number of haematopoietic cells in wild-type and *gata2a*$^{\Delta i4/\Delta i4}$ mutant WKM (Fig. 6d–f). To avoid any confounding effects in our analysis, we compared wild-type to *gata2a*$^{\Delta i4/\Delta i4}$ mutants without overt signs of infection. The *gata2a*$^{\Delta i4/\Delta i4}$ mutants showed a ~2-fold decrease in the total number of haematopoietic cells in the WKM (Fig. 6d–f). In

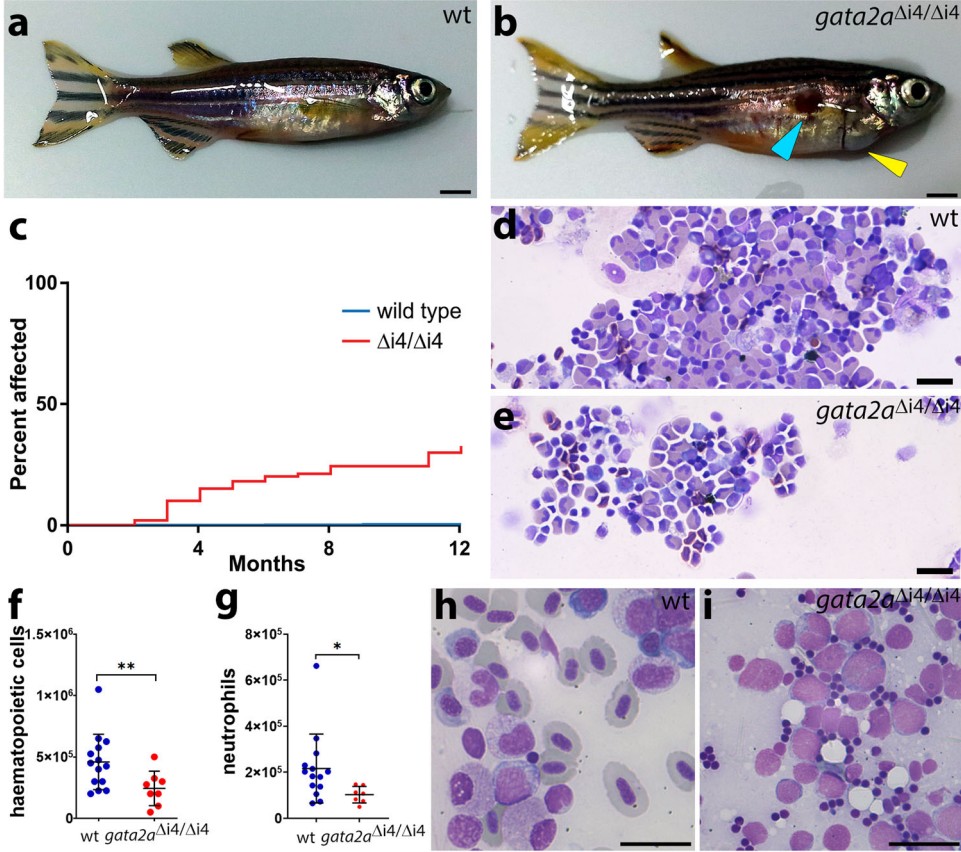

**Fig. 6 Gata2a$^{\Delta i4/\Delta i4}$ mutants show cardiac oedema, hypocellularity and marrow failure. a, b** General morphology of zebrafish adults: **a** wild-type; **b** gata2a$^{\Delta i4/\Delta i4}$ mutant showing skin infection (blue arrowhead) and pericardial oedema (yellow arrowhead). **c** Over 25% ($n = 29/108$) of gata2a$^{\Delta i4/\Delta i4}$ mutants (red) catch infections or suffer from heart oedemas by 6 months. Only around 65% ($n = 69/108$) survive for more than 12 months without overt signs of infections. Fewer than 1% ($n = 2/500$) of wild-type fish (blue) exhibit such defects. The graph does not include deaths by other causes. **d, e** May-Grunwald/Wright-Giemsa staining in cytospins of haematopoietic cells isolated from the WKM of zebrafish adults: **d** wild-type; **e** gata2a$^{\Delta i4/\Delta i4}$ mutant. Note the decrease in cell numbers. **f** Cell counts of haematopoietic cells isolated from WKM of wild-type ($n = 14$) and gata2a$^{\Delta i4/\Delta i4}$ mutants ($n = 8$). The gata2a$^{\Delta i4/\Delta i4}$ mutants show a ~2-fold decrease in haematopoietic cell numbers in the WKM ($\mu_{wt} = 4.37 \times 10^5$; $\mu_{mut} = 2.37 \times 10^5$, $p = 0.0185$, Mann-Whitney test). (g) Number of neutrophils isolated from WKM of wild-type ($n = 14$) and gata2a$^{\Delta i4/\Delta i4}$ mutants ($n = 7$). The gata2a$^{\Delta i4/\Delta i4}$ mutants show a ~2-fold decrease in neutrophil numbers in the WKM ($\mu_{wt} = 2.17 \times 10^5$; $\mu_{mut} = 1.03 \times 10^5$, $p = 0.0269$, Mann-Whitney test). Error bars: median cell number ± SD. **h, i** Kidney smears from 9 months post-fertilization adult animals were assessed. **h** Wild-type shows various stages of lineage differentiation. **i** WKM smear; 1 of 10 gata2a$^{\Delta i4/\Delta i4}$ mutants showed the presence of excess blasts with very little erythroid differentiation (98% blasts, >200 cells assessed). Scalebars: 2 mm (**a, b**) and 10 μm (**d, e, h, i**). See also Supplementary Fig.6.

addition, neutrophils were similarly reduced (Fig. 6g), another characteristic in common with MonoMAC syndrome patients[14]. Lastly, kidney marrow smears of ten 9-month old gata2a$^{\Delta i4/\Delta i4}$ mutants were assessed. One of the ten mutants showed an excess of immature myeloid blast cells in the WKM (>98%) and only minor erythrocyte differentiation (Fig. 6h, i). The presence of excess blasts is usually an indication of AML in humans. Together these data strongly suggest that the i4 enhancer is a critical driver of gata2a expression in adult haematopoietic cells. The enhancer deletion in gata2a$^{\Delta i4/\Delta i4}$ mutants leads to a hypocellular WKM and neutropenia, strongly suggestive of marrow failure, a hallmark of disease progression in Gata2 deficiency syndromes.

## Discussion

The sub-functionalisation of the Gata2 paralogues in zebrafish provided an opportunity to unpick the different roles of Gata2 in the multi-step process of definitive haematopoiesis. Here we have investigated the conservation of the Gata2 +9.5 enhancer and identified a homologous region in intron 4 of the zebrafish gata2a locus (gata2a-i4) that is not present in the gata2b locus. The zebrafish gata2a-i4 enhancer, like the mouse enhancer[18], is

sufficient to drive pan-endothelial expression of GFP and necessary for endothelial expression of gata2a (Figs. 1 and 2). We traced the activity of the i4 enhancer back to the PLM, the source of precursors of endothelium and HSCs[10]. This degree of sequence and functional conservation of the i4 enhancer led us to hypothesize that Gata2a might play a role in definitive haematopoiesis. Indeed, homozygous deletion of the i4 enhancer (gata2a$^{\Delta i4/\Delta i4}$) allowed us to uncover a previously unknown function of Gata2a in regulating the initial expression of runx1 and gata2b in HE. Although cmyb expression in HE was decreased in gata2a$^{\Delta i4/\Delta i4}$ mutants, it was more severely reduced in gata2b$^{-/-}$ mutants, suggesting that Gata2b is more important for cmyb regulation than Gata2a. However, both Gata2 orthologues regulate gene expression in the HE before the first reported EHT events at 34hpf[6].

Gata2 binds to the +9.5 enhancer to maintain its own expression in endothelial and haematopoietic cells[26,44]. In zebrafish, it is likely that Gata2a binds the GATA motifs in the i4 enhancer and loss of gata2a in the endothelium of gata2a$^{\Delta i4/\Delta i4}$ mutants (Fig. 2) seems to support this view. Interestingly, we detected a small region in intron 4 of the gata2b locus that was

not identified as a peak in our ATACseq experiment but is conserved in some fish species (Supplementary Fig. 1a) and thus could potentially represent a divergent gata2b intronic enhancer. We speculate that the positive autoregulation of Gata2 was likely retained by both gata2 orthologues in zebrafish, but this possibility remains to be investigated.

The gata2a$^{\Delta i4/\Delta i4}$ mutants recovered from the early defects in HE programming and displayed normal expression levels of cmyb in the CHT at 4dpf and rag1 in the thymus at 5dpf, used as indicators of the definitive haematopoietic programme[11]. We hypothesized that this could be due to the presence of the two homologues of Gata2 in zebrafish[20], despite Gata2a and Gata2b proteins being only 50% identical[21]. Indeed, forced expression of gata2b under the gata2a-i4 enhancer rescued DA expression of runx1 in the gata2a$^{\Delta i4/\Delta i4}$ mutants to wild-type levels and suboptimal depletion of gata2b in the gata2a$^{\Delta i4/\Delta i4}$ mutants resulted in more severe reduction in cmyb expression in the CHT by 4dpf (Fig. 4). In addition, we demonstrated that Notch signalling, a known regulator of gata2b expression[21], is sufficient to rescue the initial HE programming defect induced by deletion of the gata2a-i4 enhancer. We propose a model in which gata2a acts upstream of runx1 and gata2b independently of Notch to initiate HE programming. The regulation of gata2b by Gata2a is transient, and the timing largely coincides with the natural decrease in endothelial expression of gata2a by 30hpf. After this stage, endothelial Notch signalling takes over the regulation of runx1 and gata2b expression, acting as a fail-safe mechanism that buffers against fluctuations in the system caused by loss of one or more of the initial inputs (in this case, Gata2a).

Despite the apparent haematopoietic recovery, we observed a high incidence of infections and oedema in gata2a$^{\Delta i4/\Delta i4}$ adults, and a striking decrease in the number of haematopoietic cells in the WKM. The decrease in haematopoietic cells in particular is reminiscent of the loss of proliferative potential of haematopoietic Gata2$^{+/-}$ heterozygous cells in the mouse[17,45]. This raises the possibility that in zebrafish the gata2a and gata2b paralogues may function as two Gata2 'alleles' that together regulate the haematopoietic output of the WKM. This will be addressed by comparing the adult phenotypes of gata2a$^{\Delta i4/\Delta i4}$ and gata2b$^{-/-}$ mutants.

Taken together, our initial characterization of WKM shows that gata2a$^{\Delta i4/\Delta i4}$ mutants present a phenotype consistent with Gata2 deficiency syndromes in humans brought about by GATA2 haploinsufficiency[12,26]. Strikingly, about 10% of all MonoMAC patients show mutations in the conserved +9.5 enhancer[12,14], the corresponding regulatory element to the i4 enhancer. The i4 enhancer is active in the lymphoid + HSPC fraction that contains the HSC activity[46], in the progenitor cells and in the myeloid fraction that contains eosinophils, previously identified as expressing high levels of a gata2a-GFP BAC transgenic reporter[41]. Thus, it is likely that gata2a$^{\Delta i4/\Delta i4}$ adult fish show lineage-specific differentiation defects. Further characterization of the gata2a$^{\Delta i4/\Delta i4}$ mutants will uncover which haematopoietic cells are most affected by the loss of i4 enhancer activity and how Gata2a regulates haematopoietic output, thus establishing a zebrafish animal model for human diseases linked to Gata2 haploinsufficiency.

## Methods

**Maintenance of zebrafish.** Zebrafish (Danio rerio) were maintained in flowing system water at 28.5 °C, conductance 450–550 μS and pH 7.0 ± 0.5 as described[47]. Fish suffering from infections or heart oedemas were culled according to Schedule 1 of the Animals (Scientific Procedures) Act 1986. Eggs were collected by natural mating. Embryos were grown at 24–32 °C in E3 medium with methylene blue and staged according to morphological features corresponding to respective age in hours or days post-fertilization (hpf or dpf, respectively). Published strains used in this work were wild-type (wt$^{KCL}$), Tg(−6.0itga2b:EGFP)$^{la2}$ [38,48] and Tg(kdrl:GFP)$^{s843}$ [28];

animals were used at embryonic and larval stages and as adults (male and female) as specified in the figures. All animal experiments were approved by the relevant University of Oxford, University of Birmingham and Erasmus University ethics committees.

**ATAC-seq.** Tg(kdrl:GFP)$^{s843}$ embryos were dissociated for FACS at 26-27hpf to collect kdrl$^+$ and kdrl$^-$ cell populations (40,000–50,000 cells each). They were processed for ATAC library preparation using optimised standard protocol[27]. Briefly, after sorting into Hanks' solution (1xHBSS, 0.25% BSA, 10 mM HEPES pH8), the cells were spun down at 500 g at 4 °C, washed with ice-cold PBS and resuspended in 50 μl cold Lysis Buffer (10 mM Tris-HCl, 10 mM NaCl, 3 mM MgCl$_2$, 0.1% IGEPAL, pH 7.4). The nuclei were pelleted for 10 min. at $500 \times g$ at 4 °C and resuspended in the TD Buffer with Tn5 Transposase (Illumina), scaling the amounts of reagents accordingly to the number of sorted cells. The transposition reaction lasted 30 min. at 37 °C. The DNA was purified with PCR Purification MinElute Kit (QIAGEN). In parallel, transposase-untreated genomic DNA from kdrl$^+$ cells was purified with the DNeasy® Blood & Tissue Kit (QIAGEN). The samples were amplified with appropriate Customized Nextera primers[27] in NEB-Next High-Fidelity 2x PCR Master Mix (NEB). The libraries were purified with PCR Purification MinElute Kit (QIAGEN) and Agencourt AMPure XP beads (Beckmann Coulter). The quality of each library was verified using D1000 ScreenTape System (Agilent). Four biological replicas of the libraries were quantified with the KAPA Library Quantification Kit for Illumina® platforms (KAPA Biosystems). The libraries were pooled (including the Tn5-untreated control), diluted to 1 ng/μl and sequenced using 75 bp paired-end reads on Illumina HiSeq 4000 (Wellcome Trust Centre for Human Genetics, Oxford). Raw sequenced reads were checked for base qualities, trimmed where 20% of the bases were below quality score 20, and filtered to exclude adapters using Trimmomatic (Version 0.32) and mapped to Zv9 reference genome (comprising 14,612 genes)[49] using BWA with default parameters. The results were visualised using UCSC Genome Browser (http://genome-euro.ucsc.edu/)[50]. The eight data sets were analysed with Principal Component Analysis (PCA) to identify outliers. Correlation among kdrl:GFP$^+$ and kdrl:GFP$^-$ samples was assessed with a tree map. The peaks were called for each sample using the Tn5-untreated control as input. We identified the common peaks between replicates and then used DiffBind (EdgeR method) to identify differential peaks between kdrl:GFP$^+$ and kdrl:GFP$^-$ samples (Supplementary Data 2). The threshold for differential peaks was $p < 0.05$.

**Generation of transgenic and mutant zebrafish lines.** Genomic regions containing the identified 150bp-long gata2a-i4 enhancer flanked by ±500 bp (i4–1.1 kb) or ±150 bp (i4–450bp) were amplified from wild-type zebrafish genomic DNA with NEB Phusion® polymerase (see Supplementary Table 1 for primer sequences) and cloned upstream of an E1b minimal promoter and GFP into a Tol2 recombination vector (Addgene plasmid #37845[51]) with Gateway® cloning technology (Life Technologies™) following the manufacturer's protocol. One-cell zebrafish embryos were injected with 1 nl of an injection mix, containing 50 pg gata2a-i4-E1b-GFP-Tol2 construct DNA + 30 pg tol2 transposase mRNA[31]. Transgenic founders (Tg (gata2a-i4-1.1 kb:GFP) and (gata2a-i4-450 bp:GFP)) were selected under a wide-field fluorescent microscope and outbred to wild-type fish. Carriers of monoallelic insertions were detected by the Mendelian distribution of 50% fluorescent offspring coming from wild-type outcrosses. These transgenics were then inbred to homozygosity.

To generate the i4 deletion mutant, we identified potential sgRNA target sites flanking the 150 bp conserved region within intron 4 of the gata2a locus (see Fig. 1a, Supplementary Fig 3a). sgRNAs were designed with the CRISPR design tool (http://crispr.mit.edu/, see Supplementary Table 1 for sequences) and prepared as described[32]. To reduce potential off-target effects of CRISPR/Cas9, we utilized the D10A 'nickase' version of Cas9 nuclease[52,53], together with two pairs of sgRNAs flanking the enhancer (Supplementary Table 1, Supplementary Fig. 3a, a). We isolated two mutant alleles with deletions of 215 bp (Δ78–292) and 231 bp (Δ73–303) (Supplementary Fig. 3b). Both deletions included the highly conserved E-box, Ets and GATA transcription factor binding sites (Supplementary Fig. 3b). The Δ73–303 allele was selected for further experiments and named Δi4. Adult zebrafish were viable and fertile as heterozygous (gata2a$^{\Delta i4/+}$) or homozygous (gata2a$^{\Delta i4/\Delta i4}$). To unambiguously genotype wild types, heterozygotes and homozygous mutants, we designed a strategy consisting of two PCR primer pairs (Supplementary Fig. 3a, c). One primer pair flanked the whole region, producing a 600 bp wild-type band and 369 bp mutant band. In the second primer pair, one of the primers was designed to bind within the deleted region, only giving a 367 bp band in the presence of the wild-type allele (Supplementary Fig. 3c).

To generate the gata2b mutant we designed a CRISPR/Cas9 strategy for a frameshift truncating mutant in exon 3 deleting both zinc fingers. sgRNAs were designed as described above and guides were prepared according to Gagnon et al.[54] with minor adjustments. Guide RNAs were generated using the Agilent SureGuide gRNA Synthesis Kit, Cat# 5190–7706. Cas9 protein (IDT) and guide were allowed to form ribonucleoprotein structures (RNPs) at RT and injected in 1 cell stage oocytes. 8 embryos were selected at 24 hpf and lysed for DNA isolation. Heteroduplex PCR analysis was performed to test guide functionality and the other embryos from the injection were allowed to grow up. To aid future genotyping we selected mutants by screening F1 for a PCR detectable integration or deletion in

exon 3. Sequence verification showed that founder 3 had a 28 nt integration resulting in a frameshift truncating mutation leading to 3 new STOP codons in the third exon. To get rid of additional mutations caused by potential off-target effects, founder 3 was crossed to WT for at least 3 generations. All experiments were performed with offspring of founder 3.

**Fluorescence-activated cell sorting (FACS).** Approximately 100 embryos at the required stage were collected in Low Binding® SafeSeal® Microcentrifuge Tubes (Sorenson) and pre-homogenized by pipetting up and down in 500 µl Deyolking Buffer (116 mM NaCl, 2.9 mM KCl, 5 mM HEPES, 1 mM EDTA). They were spun down for 1 min. at 500 g and incubated for 15 min. at 30 °C in Trypsin + Collagenase Solution (1xHBSS, 0.05% Gibco® Trypsin + EDTA (Life Technologies™), 20 mg/ml collagenase (Sigma)). During that time, they were homogenized by pipetting up and down every 3 min. The lysis was stopped by adding 50 µl foetal bovine serum and 650 µl filter-sterilized Hanks' solution (1xHBSS, 0.25% BSA, 10 mM HEPES pH8). The cells were rinsed with 1 ml Hanks' solution and passed through a 40 µm cell strainer (Falcon®). They were resuspended in ~400 µl Hanks' solution with 1:10,000 Hoechst 33258 (Molecular Probes®) and transferred to a 5 ml polystyrene round bottom tube for FACS sorting. The cells were sorted on FACSAria Fusion sorter by Kevin Clark (MRC WIMM FACS Facility). The gates of GFP (488–530) and DsRed (561–582) channels were set with reference to samples derived from non-transgenic embryos. The fluorescence readouts were compensated when necessary. For ATAC-seq library preparation, the cells were sorted into Hank's solution. For RNA isolation, the cells were sorted directly into RLT Plus buffer (QIAGEN) + 1% β-mercaptoethanol and processed with the RNEasy® Micro Plus kit (QIAGEN), according to the accompanying protocol. The RNA was quantified and its quality assessed with the use of Agilent RNA 6000 Pico kit. All RNA samples were stored at −80 °C.

**SYBR® Green qRT-PCR.** 3 µl of the cDNA diluted in $H_2O$ were used for technical triplicate qRT-PCR reactions of 20 µl containing the Fast SYBR® Green Master Mix (Thermo Fisher Scientific) and appropriate primer pair (see Supplementary Table 1). The reactions were run on 7500 Fast Real-Time PCR System (Applied Biosystems) and the results were analysed with the accompanying software. No-template controls were run on each plate for each primer pair. Each reaction was validated with the melt curve analysis. The baseline values were calculated automatically for each reaction. The threshold values were manually set to be equal for all the reactions run on one plate, within the linear phase of exponential amplification. The relative mRNA levels in each sample were calculated by subtracting the geometric mean of Ct values for housekeeping genes $eef1a1l1$ and $ubc$ from the average Ct values of the technical triplicates for each gene of interest. This value (ΔCt) was then converted to a ratio relative to the housekeeping genes with the formula $2^{-\Delta Ct}$.

**Fluidigm Biomark qRT-PCR.** To quantify the differences in $gata2a$ expression between wild-type and mutant ECs, we crossed homozygous $gata2a^{\Delta i4/\Delta i4}$ mutants to Tg($kdrl$:GFP) transgenics to generate Tg($kdrl$:GFP); $gata2a^{\Delta i4/\Delta i4}$ embryos. These fish, along with non-mutant Tg($kdrl$:GFP), were used for FACS-mediated isolation of $kdrl$:GFP$^+$ and $kdrl$:GFP$^-$ cells to quantitatively compare mRNA expression levels of $gata2a$ in the endothelial and non-endothelial cells of wild-type and $gata2a^{\Delta i4/\Delta i4}$ embryos, using the Fluidigm Biomark™ qRT-PCR platform. Briefly, 1 ng RNA from FACS-sorted cells was used for Specific Target Amplification in a 10 µl reaction with the following reagents: 5 µl 2xBuffer and 1.2 µl enzyme mix from SuperScript III One-Step Kit (Thermo Fisher Scientific), 0.1 µl SUPERase• In™ RNase Inhibitor (Ambion), 1.2 µl TE buffer (Invitrogen), 2.5 µl 0.2x TaqMan® assay mix (see Supplementary Table 2 for the details of TaqMan® assays). The reaction was incubated for 15 min. at 50 °C, for 2 min. at 95 °C and amplified for 20 cycles of 15 s at 95 °C/4 min. at 60 °C. The cDNA was diluted 1:5 in TE buffer and stored at −20 °C. Diluted cDNA was used for qRT-PCR according to the Fluidigm protocol for Gene Expression with the 48.48 IFC Using Standard TaqMan® Assays (Supplementary Table 2). Each sample was run in 3–4 biological replicates. The collected data were analysed with Fluidigm Real-Time PCR Analysis software (version 4.1.3). The baseline was automatically corrected using the built-in Linear Baseline Correction. The thresholds were manually adjusted for each gene to fall within the linear phase of exponential amplification, after which they were set to equal values for the housekeeping genes: $rplp0$, $rpl13a$, $cops2$[55], $lsm12b$[56] and $eef1a1l1$. The relative mRNA levels for each sample were calculated by subtracting the geometric mean of Ct values for the housekeeping genes from the Ct value for each gene of interest. This value (ΔCt) was then converted to a ratio relative to the housekeeping genes with the formula $2^{-\Delta Ct}$. The ΔCt values were analysed with 2-tailed paired-samples t-tests with 95% confidence levels.

**Flow cytometry and isolation of WKM haematopoietic cells.** Single cell suspensions of WKM cells were prepared from adult zebrafish kidneys of the required genotypes as described[57]. Briefly, adult zebrafish were first euthanized in 0.5% tricaine in PBS and dissected to remove the kidney. WKM cells were recovered by vigorous pipetting in 0.5 ml PBS with 10% Foetal Calf Serum (PBS + 10%FCS), followed by filtration in a cell strainer (FALCON, ref 352235) pre-coated with PBS + 10%FCS. Strainers were rinsed with PBS + 10%FCS and the cells spun down

(~300 g, 10 min at 4 °C) and ressuspended in 200–500 µl PBS + 10%FCS with Hoechst 33342 1:10000 (Hoechst 33342, H3570, ThermoScientific). Flow cytometry analysis was performed on a FACS Aria II (BD Biosciences) after exclusion of dead cells by uptake of Hoechst dye, as described[41]. WKM cell counts were performed on a PENTRA ES60 (Hariba Medical) following the manufacturer's instructions. Note that the cell counter does not recognize the zebrafish nucleated erythrocytes, so these were excluded from this analysis. Cell counts for each genotype were analysed with 2-tailed paired-samples t-tests with 95% confidence levels, using a Mann-Whitney test for non-parametric distribution. The scatter plots were generated using GraphPad Prism 8.0 and show medians ± SD.

**May-Grunwald and Wright-Giemsa staining.** Cell staining with May-Grunwald (MG) stain (Sigma MG500) and Giemsa (GIEMSA STAIN, FLUKA 48900) was performed on haematopoietic cell samples. After cytospin, slides are allowed to air-dry and were stained for 5 min at room temperature with a 1:1 mix of MG:distilled water. Next, slides were drained and stained with a 1:9 dilution of Giemsa:distilled water solution for 30 min at room temperature. Excess solution was drained and removed by further washes in distilled water. Finally, the slides were air-dried and mounted in DPX (06522, Sigma) for imaging.

**Whole-mount in situ hybridization and immunohistochemistry.** Whole-mount in situ hybridization (ISH) was carried out as described previously[58], using probes for $kdrl$, $runx1$, $cmyb$, $gata2a$, $gata2b$, $rag1$[4,37,59,60] and $gfp$ (Supplementary Table 1). For conventional ISH embryos were processed, imaged and the ISH signal quantified as described[34]. Briefly, the pixel intensity values were assessed for normal distribution with a Q-Q plot and transformed when necessary. Mean values (µ) of each experimental group were analysed with 2-tailed independent-samples t-tests or with ANOVA with 95% confidence levels, testing for the equality of variances with a Levene's or Brown-Forsythe test and applying the Welch correction when necessary. For ANOVA, differences between each two groups were assessed with either Tukey's post-hoc test (for equal variances) or with Games-Howell test (for unequal variances). In all these analysis, the IBM® SPSS® Statistics (version 22) or GraphPad Prism 8.0 package were used.

For the analysis of $cmyb$ expression in the CHT at 4dpf, the embryos scored as 'high' or 'low' were tested for equal distribution between morphants and uninjected controls or among wild-type, heterozygous and mutant genotypes with contingency Chi-squared tests, applying Continuity Correction for 2 × 2 tables, using IBM® SPSS® Statistics (version 22).

For fluorescent ISH (FISH) combined with immunohistochemistry, ISH was performed first following the general whole-mount in situ hybridisation protocol. The signal was developed with SIGMAFAST Fast Red TR/Naphthol, the embryos rinsed in phosphate-buffered saline with tween20 (PBT) and directly processed for immunohistochemistry. Embryos were blocked in blocking buffer (5% goat serum/ 0.3% Triton X-100 in PBT) for 1 h at RT before incubated with primary antibody against GFP (rabbit, 1:500, Molecular Probes), diluted in blocking buffer overnight at 4 °C. Secondary antibody raised in goat coupled to AlexaFluor488 (Invitrogen) was used in 1:500 dilutions for 3 h at RT. Hoechst 33342 was used as a nuclear counterstain.

Fluorescent images were taken on a Zeiss LSM880 confocal microscope using ×40 or ×63 oil immersion objectives. Images were processed using the ZEN software (Zeiss).

**Fluorescence microscopy and cell counting.** For widefield fluorescence microscopy, live embryos were anaesthetised with 160 µg/ml MS222 and mounted in 3% methylcellulose and imaged on a AxioLumar V.12 stereomicroscope (Zeiss) equipped with a Zeiss AxioCam MrM. To count $itga2b$-GFP$^{high}$ and $itga2b$-GFP$^{low}$ cells in the CHT, Tg($itga2b$:GFP;$kdrl$:mCherry); $gata2a^{\Delta i4/+}$ animals were incrossed and grown in E3 medium supplemented with PTU to prevent pigment formation. At 5dpf, the larvae were anaesthetised with MS222 and the tail was cut and fixed for 1 h at room temperature in 4% PFA. Next, the tails were mounted on 35 mm glass bottomed dishes (MAtTEK) in 1% low melt agarose and imaged using a ×40 oil objective on an LSM880 confocal microscope (Zeiss). Cells in the CHT region were counted manually on Z-stacks as '$itga2b$:GFP$^{low}$' (HSPCs) or '$itga2b$: GFP$^{high}$' (thrombocytes). Genomic DNA from the heads was extracted and used for genotyping as described above. Cell counts for each genotype were analysed with 2-tailed paired-samples t-tests with 95% confidence levels, using a Mann-Whitney test for non-parametric distribution. The graphs were generated using GraphPad Prism 8.3.0 and show medians ± SD.

**Statistics and reproducibility.** Data were analysed using either IBM® SPSS® Statistics (version 22) or GraphPad Prism software (v8.02). In situ quantification data was analysed with 2-tailed independent-samples t-tests or with ANOVA with 95% confidence levels, testing for the equality of variances with a Levene's or Brown-Forsythe test and applying the Welch correction when necessary. For ANOVA, differences between each two groups were assessed with either Tukey's post-hoc test (for equal variances) or with Games-Howell test (for unequal variances). Alternatively, data were analysed with an appropriate non-parametric test (Kruskall-Wallis) followed by Dunn's multiple comparisons test or uncorrected Dunn's test where appropriate. Cell count data were analysed with 2-tailed paired-

samples t-tests with 95% confidence levels, using a Mann-Whitney test for non-parametric distribution. Gene expression data were analysed with 2-tailed paired-samples t-tests with 95% confidence levels.

All experiments were repeated at least three times with similar results obtained; sample sizes are shown in the respective figure legends.

**Reporting summary**. Further information on research design is available in the Nature Research Reporting Summary linked to this article.

## Data availability

All data generated or analyzed during this study (images, quantitation data in the form of graphs and ATACseq data) are included in this published article and its supplementary information files. The source data are available in Supplementary Data 1 and the list of the called ATACseq peaks is available in Supplementary Data 2. The ATACseq data was deposited in GEO (Accession number GSE143763).

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

## Acknowledgements

We thank the staff of the Biomedical Services Units (Oxford, Birmingham and Rotterdam) for fish husbandry. We thank Kevin Clark and Sally-Ann Clark from the WIMM flow cytometry facility for cell sorting. The flow cytometry facility is supported by the MRC HIU, MRC MHU (MC_UU_12009), NIHR Oxford BRC and John Fell Fund (131/030 and 101/517), the EPA fund (CF182 and CF170). We thank the Wolfson Imaging Centre Oxford for imaging. The Centre is supported by a Wolfson Foundation (grant 18272), and an MRC/BBSRC/EPSRC grant (MR/K015777X/1) to MICA – Nanoscopy Oxford. Both facilities were supported by the WIMM Strategic Alliance awards G0902418 and MC_UU_12025. We thank Fatma Kok and Douglas Vernimmen for critical reading of the manuscript. We thank Feng Liu for the generous gift of the *fli1a*-NICD:GFP construct. This research was supported by the British Heart Foundation (BHF Oxford CoRE and BHF IBSR Fellowship FS/13/50/30436 to R.M. and M.K.), by a Wellcome Trust Chromosome and Developmental Biology PhD Scholarship (#WT102345/Z/13/Z. to T.D.) and by the MRC MHU programme (MC_UU_12009/8 to R.P.).

## Author contributions

T.D., M.K., C.K., E.P. and R.M. designed the study. T.D., M.K., C.K., J.P.-Z., K.G., C.B.M. and R.M. performed experiments and analyzed the data. J.P-Z., B.F. and K.G. performed experiments. R.R. performed the bioinformatics analyses, T.D. and R.M. wrote the paper and R.P., E.P. and R.M. edited the paper. R.P., E.P. and R.M. secured funding.

## Competing interests

The authors declare no competing interests.
