## [Peer Review File · Communications Biology]

Reviewers' comments:

Reviewer #1 (Remarks to the Author):

The manuscript by Dobrzycki et al. describes studies on the function of a GATA2 intronic enhancer in zebrafish that has previously been studied extensively in mouse systems and to a lesser extent in humans. Overall, data quality is high and analyzing this essential enhancer in additional systems has potential to yield significant new insights and/or confirm existing concepts that have emerged from studies in mouse and man. Since mutations of this enhancer in humans (like GATA2 coding mutations) create a predisposition to develop immunodeficiency, bone marrow failure and leukemia, the zebrafish model is of considerable interest. However, throughout the manuscript there are issues with the presentation, specifically related to the mutation causing AML and relating the fish phenotypes to the complex human disorder MonoMAC. MonoMAC involves complex perturbations of the immune system, which differs grossly between man and fish, and I recommend eliminating the presentation of the results in the context of a MonoMAC model. This is unnecessary, as the results are interesting and represent a significant contribution to the literature without having to make the stretch that the loss of GATA2 expression, bone marrow failure etc mimics human MonoMAC.

Additional Comments:

1. As noted above, based on the data presented, I recommend eliminating MonoMAC from the title and also statements in the text indicating that the mutant fish model human MonoMAC. Based on this complex human disorder, I do not see strong evidence that the fish phenotypes "model" this disease.
2. In many prior studies, the mouse enhancer was deemed the +9.5 enhancer, based on its localization relative to the promoter. The precise relationship of the intron 4 sequences in zebrafish to mouse and man is unclear from the data presented. Fig. 1A text is very small and fuzzy, and I cannot read this. Some of the in situ hybridization analyses are also quite small and fuzzy, and I encourage the authors to present larger figures.
3. Fig. 3 – The authors concluded that GATA2a is required upstream of GATA2b and RUNX1. Gao et al. (JEM 2013) had already described a mechanism in which murine GATA2 functions upstream of RUNX1 in the AGM. This is an established concept.
4. Fig. 6 – The data of panel I is very interesting. Is this really "AML"? A more thorough presentation of evidence would be required to refer to this as AML.
5. Why relegate the single cell transcriptomic analysis to supplement? This has potential to be an important contribution to the study.

Reviewer #2 (Remarks to the Author):

In this manuscript Dobrzycki et al describe the role of gata2a enhancer (i4) in hemogenic endothelial (HE) specification. Although the role of this Gata2 enhancer has been reported in other systems, they show, for the first time in zebrafish, that its deletion reduces the expression of gata2a specifically in endothelial cells. They prove that endothelial gata2a expression modulated by the i4 enhancer is necessary for the initiation of expression of gata2b and subsequently runx1, and thus HSPC

specification. However, this defect is compensated after 48hpf. They suggest that *gata2b*, the expression of which is necessary for HE specification, can be regulated by two independent factors, one based on *gata2a* expression driven by the *i4* enhancer and Notch signaling. Indeed, upon notch inhibition, *runx1* expression is reduced at a later stage (36 hpf). They propose that Notch signaling is responsible for the observed phenotype compensation after 48 hpf. Finally, they show that adult zebrafish in which the *gata2a* *i4* enhancer has been deleted, accumulate features that resemble those of MonoMAC syndrome patients, which is significant, as this fish can serve as a model for the above-mentioned syndrome.

This is an exciting and very thorough paper. The authors have done a good job proving their points and supporting their conclusions with a satisfactory amount of data, both quantity- and quality-wise. However, the manuscript can be further improved.

Major points

- The authors conclude that notch is an additional input on *gata2b* expression and responsible for rescuing the hematopoietic development at later stages by upregulating the expression of *gata2b*. To support this argument, they inhibit notch signaling in *gata2aΔi4/Δi4* and find that upon this inhibition they can detect reduced expression of *runx1* even at later stages. However, even if it is known that *gata2b* expression is regulated by notch, it would be nice if they could show this. Could they repeat the same experiment and perform an in situ hybridization for *gata2b* to really prove that notch inhibition prevents its upregulation? Can activation of notch specifically in endothelial/HE cells prevent downregulation of *gata2b* and *runx1* even at early stages (e.g. 25hpf) in *gata2aΔi4/Δi4* zebrafish embryos? The authors speculate that activation of notch is sufficient to rescue the phenotype, as they mention clearly in their discussion, but they should do the actual experiment.
- The authors do several experiments with the *gata2a-i4* enhancer and they have identified a minimal sequence of 150bp that can actually drive *gata2a* expression in endothelial cells. Can the authors perform some mutations on transcription factor motifs to identify the factors that regulate the activity of this enhancer?
- The authors detected AML in one of their three mutants. This is very interesting. Could they use more adult fish and observe what is the propensity of this? It would be great if the authors could prove that these blasts that they observe are transplantable.

Minor points

- In the abstract (p.2 line 24) the authors claim that their mutants developed acute myeloid leukemia. This is a very strong statement that should be softened to comply with the actual data.
- P6 line 124. The statement in this phrase is wrong. ATAC-seq provides regions of open and not necessarily active chromatin.
- Why did the authors perform the motif analysis only on enhancer regions? It is well established that GATA factors bind primarily to enhancers. Can the authors expand their motif search?
- The authors claim that four peaks were differentially accessible in the *gata2a* locus. Can the authors provide any information on the other peaks? Why did they choose the *i4* enhancer?
- The authors show that their mutants have reduced *gata2b* expression. Can this result be recapitulated by simply knocking down *gata2a*?
- The authors should provide some more statistics. For example, on figure 5 they use a sub-optimal amount of *gata2b* morpholino, but in their quantification they do not present the statistics between wild type and wild type injected with the morpholino. Similarly and on the same figure, they show that ectopic expression of *gata2b* increases *runx1* expression in *gata2aΔi4/Δi4* mutants. They also claim this rescue reaches the levels of *runx1* expression in wild type fish, but this statistic comparison is missing from the figure. Is really the difference non-significant? Finally, on Figure S5, they claim that the absence of *gata2b* or both *gata2a* and *gata2b* has the same impact on *cmyb* expression at 33 hpf.

Could they present the statistic comparison between these two and show it is not significant?

- Although they are quite meticulous describing their experiments and the numbers of embryos used in their figure legends, the authors should also include numbers in all their zebrafish experiments (e.g. in situ hybridization experiments). For instance, these numbers are missing on figure 5A,C,F and supplementary figure 5B-D.
- Although no doubt is cast on the validity of the results, the number of embryos the authors present on pictures of in situ hybridization is relatively low. Could they examine a bit more embryos (at least regarding their basic phenotypes in *gata2aΔi4/Δi4* mutants) and include them on the figures?
- Could the authors check whether (except for *kdrl*) the expression of the arterial marker *ephrinβ2* changes in *gata2aΔi4/Δi4* mutants?
- On figure 4G,H the embryos are, according to the manuscript, 5dpf embryos, but instead they are labelled as 4dpf.

Reviewer #3 (Remarks to the Author):

This manuscript shows the exquisite regulation of hematopoietic development by the 2 Gata2 genes (2a and 2b) in Zebrafish embryos. The authors describe the conserved enhancer in the 4th intron of Gata2a and carefully manipulate it through Crispr/Cas9 methodology. These embryos are examined at several (mainly two) time points and effects on endothelial cell, hemogenic endothelial and hematopoietic cell development are investigated. Because the *Kdrl* GFP reporter is used, endothelial cells can be sorted and expression analyses carried out. Moreover, combinations of transcriptome and protein expression analyses on zebrafish embryos validates and strengthens the data presented. The authors clearly show that the two Gata2s act at different stages of development and that Gata2b is regulated by two different inputs - by Gata2a and Notch. Finally, they suggest that Gata2a is required in adult hematopoiesis and that its deficiency presents a phenotype very much like Gata2 haploinsufficiency in the human MonoMac syndrome. Altogether the studies are well-performed, statistical analysis is appropriate and provide novel findings that advance the field. It presents important new insights into the complexity of Gata2 expression and function that previously have been difficult to reconcile in studies of mouse and human Gata2.

Minor comment

Some data figure legends do not contain information on number of cells examined.

Reviewers' comments:

Reviewer #1 (Remarks to the Author):

The manuscript by Dobrzycki et al. describes studies on the function of a GATA2 intronic enhancer in zebrafish that has previously been studied extensively in mouse systems and to a lesser extent in humans. Overall, data quality is high and analyzing this essential enhancer in additional systems has potential to yield significant new insights and/or confirm existing concepts that have emerged from studies in mouse and man. Since mutations of this enhancer in humans (like GATA2 coding mutations) create a predisposition to develop immunodeficiency, bone marrow failure and leukemia, the zebrafish model is of considerable interest. However, throughout the manuscript there are issues with the presentation, specifically related to the mutation causing AML and relating the fish phenotypes to the complex human disorder MonoMAC. MonoMAC involves complex perturbations of the immune system, which differs grossly between man and fish, and I recommend eliminating the presentation of the results in the context of a MonoMAC model. This is unnecessary, as the results are interesting and represent a significant contribution to the literature without having to make the stretch that the loss of GATA2 expression, bone marrow failure etc mimics human MonoMAC.

We thank the reviewer for their comments and are pleased that they find our study to be of high quality and significance. We have taken the comments on board and removed the focus from claims that our model mimics MonoMAC syndrome.

Additional Comments:

1. As noted above, based on the data presented, I recommend eliminating MonoMAC from the title and also statements in the text indicating that the mutant fish model human MonoMAC. Based on this complex human disorder, I do not see strong evidence that the fish phenotypes “model” this disease.

We agree that MonoMAC syndrome is indeed a very complex human disorder, with varied phenotypical presentation and penetrance (see for example, Donadieu et al, 2018, *Haematologica*). Whether the *gata2a* mutant is a faithful MonoMAC syndrome model is the subject of ongoing work in our lab. We have now modified the title and text throughout the manuscript to address the reviewer’s concern on the appropriateness of our mutant to model MonoMAC syndrome.

2. In many prior studies, the mouse enhancer was deemed the +9.5 enhancer, based on its localization relative to the promoter. The precise relationship of the intron 4 sequences in zebrafish to mouse and man is unclear from the data

presented. Fig. 1A text is very small and fuzzy, and I cannot read this. Some of the in situ hybridization analyses are also quite small and fuzzy, and I encourage the authors to present larger figures.

The i4 enhancer is located in intron 4, in the position that corresponds to the +9.5Kb enhancer in mouse and human. In the previous figure, we used the sequence conservation tracks as they normally appear in the widely used UCSC browser. We have now modified Figure 1 by splitting panel A into two panels (see new Figure 1). We modified panel B to highlight better the sequence conservation at the *gata2a* locus between zebrafish and the other organisms, including mouse and human. We also increased font size and simplified the illustration in panel B to make it easier to read.

We thank the reviewer for pointing out the issue with image quality. The lower quality of the figures is likely due to the figures being compressed to suit a PDF format for submission. We will make sure to provide the journal with high quality figures in the resubmitted version.

3. Fig. 3 – The authors concluded that GATA2a is required upstream of GATA2b and RUNX1. Gao et al. (JEM 2013) had already described a mechanism in which murine GATA2 functions upstream of RUNX1 in the AGM. This is an established concept. We agree with the reviewer that regulation of Runx1 by Gata2 is indeed an established concept. However, because of a partial genome duplication, zebrafish have two orthologues of GATA2 – Gata2a and Gata2b – and only the latter was thought to regulate *runx1* expression in zebrafish (see Butko et al, 2015). Here we establish the regulatory hierarchy between these transcription factors in the zebrafish and present evidence suggesting that early expression of *runx1* and *gata2b* is driven by Gata2a, whereas later expression (i.e. after 30hpf) is mainly driven by Notch signalling.

We now show that expression of *runx1* is already decreased at 24hpf in *gata2a*^{Δi4/Δi4} mutants (new Supp fig 4A-C) a difference still detectable at 36hpf. By contrast, *gata2b* expression was indistinguishable between wild type and in *gata2a*^{Δi4/Δi4} mutants by 30hpf (new Supplementary Fig 4E). Altogether, these data suggest that *runx1* and *gata2b* expression in HE is initially regulated by Gata2a. In addition, our new experiments (new Supplementary Fig. 5) show that Notch signalling is indeed a regulator of *runx1* (and *gata2b*), but independently of Gata2a.

4. Fig. 6 – The data of panel I is very interesting. Is this really “AML”? A more thorough presentation of evidence would be required to refer to this as AML. We thank the reviewer for the comment. Indeed, the criteria to establish whether the blast-like cells we see in the WKM of *gata2a*^{Δi4/Δi4} mutants are AML includes other parameters including transplantability. We have now expanded our WKM morphology analysis in ten *gata2a* mutants and found no additional animals showing

excess blasts in the WKM. This low penetrance of the AML-like phenotype precludes us from performing transplantation experiments reliably to assess whether indeed the excess blasts are actually AML or not. Therefore, we have modified the text in the results, abstract and discussion to reflect that although the presence of blasts in bone marrow biopsies is usually an indication of AML in humans, we do not have enough experimental evidence that the blast-like cells we found in the *gata2a*^{Δi4/Δi4} mutants are indicative of AML.

5. Why relegate the single cell transcriptomic analysis to supplement? This has potential to be an important contribution to the study.

The single cell expression data is indeed a very useful resource, generated in Ana Cvejic's lab (Athanasiadis et al, 2017) and publicly available (<https://www.sanger.ac.uk/science/tools/basicz/basicz/>). We have interrogated their data (properly referenced in the legend of supplementary Fig 6) to find which WKM cells expressed the zebrafish *gata2* orthologues and compare to the GFP reporter expression in the *gata2a* i4–GFP transgenics. Because we did not produce the original scRNAseq data and this particular comparison was mostly confirmatory, we felt it more appropriate to present it as supplementary data rather than a main figure.

Reviewer #2 (Remarks to the Author):

In this manuscript Dobrzycki et al describe the role of gata2 α enhancer (i4) in hemogenic endothelial (HE) specification. Although the role of this Gata2 enhancer has been reported in other systems, they show, for the first time in zebrafish, that its deletion reduces the expression of gata2 α specifically in endothelial cells. They prove that endothelial gata2 α expression modulated by the i4 enhancer is necessary for the initiation of expression of gata2b and subsequently runx1, and thus HSPC specification. However, this defect is compensated after 48hpf. They suggest that gata2b, the expression of which is necessary for HE specification, can be regulated by two independent factors, one based on gata2a expression driven by the i4 enhancer and Notch signaling. Indeed, upon notch inhibition, runx1 expression is reduced at a later stage (36 hpf). They propose that Notch signaling is responsible for the observed phenotype compensation after 48 hpf. Finally, they show that adult zebrafish in which the gata2 α i4 enhancer has been deleted, accumulate features that resemble those of MonoMAC syndrome patients, which is significant, as this fish can serve as a model for the above-mentioned syndrome.

This is an exciting and very thorough paper. The authors have done a good job proving their points and supporting their conclusions with a satisfactory amount of data, both quantity- and quality-wise. However, the manuscript can be further improved.

We thank the reviewer for their comments and are pleased that they find our data to be both exciting and high quality. We believe that the extra experiments suggested by the reviewers have substantially improved the manuscript and our understanding of the relationships between Gata2a, Gata2b, Runx1 and Notch signalling in the establishment of HE and HSPC programming.

Major points

- The authors conclude that notch is an additional input on gata2b expression and responsible for rescuing the hematopoietic development at later stages by upregulating the expression of gata2b. To support this argument, they inhibit notch signaling in gata2 α Δ i4/ Δ i4 and find that upon this inhibition they can detect reduced expression of runx1 even at later stages. However, even if it is known that gata2b expression is regulated by notch, it would be nice if they could show this. Could they repeat the same experiment and perform an in situ hybridization for gata2b to really prove that notch inhibition prevents its upregulation? Can activation of notch specifically in endothelial/HE cells prevent downregulation of gata2b and runx1 even at early stages (e.g. 25hpf) in gata2 α Δ i4/ Δ i4 zebrafish embryos? The authors speculate that activation of notch is sufficient to rescue the phenotype, as they mention clearly in their discussion, but they should do the actual experiment.

We thank the reviewer for suggesting these extra experiments; they have allowed us to better establish the relationships between *runx1*, *gata2a*, *gata2b* and Notch signalling. We have performed DAPM inhibitor titration experiments and show that indeed *gata2b* expression requires Notch signalling (new Supplementary Fig 5B). Low levels of the Notch inhibitor (25 μ M DAPM) are sufficient to reduce *gata2b* expression slightly, but not *runx1*. Higher doses of inhibitor (50 and 100 μ M) efficiently inhibited the expression of both genes. These data are presented in the new Supplementary Fig 5A-B and in the results section and confirms the Notch-*gata2b* regulatory link. The fact that *gata2b*, but not *runx1* expression was reduced upon treatment with 25 μ M DAPM supports the notion that the Gata2a, not Gata2b, is the main GATA factor regulating the initial expression of *runx1* in HE.

We have also analysed *gata2b* expression in *gata2a* ^{Δ i4/ Δ i4} mutants at 26hpf and 30hpf (in response to one of the minor points raised by the reviewer) and we confirmed our previous quantification and qPCR results showing that *gata2b* expression recovered by 30hpf. At 36hpf, *gata2b* expression in 25 μ M DAPM-treated embryos showed no difference between genotypes and the only robust effect that we detected was due to the Notch inhibition at 100 μ M (new Supplementary Fig. 5C). Taken together, these experiments confirm the requirement for Notch signalling for *runx1* and *gata2b* expression and have been added to new the results section.

Following the reviewer's suggestion, we have also addressed whether endothelial-specific overexpression of NICD (constitutively active Notch signalling) could rescue the decrease in *gata2b* and *runx1* found in *gata2a* ^{Δ i4/ Δ i4} mutants by using a *fli1a*-NICD:GFP construct (Liu et al, 2019, *Nature comms*). Initial test showed that expression of the construct in wildtype embryos induced a significant increase in both *gata2b* and *runx1* expression at 30hpf (new Supplementary Fig 5D,E). At 26hpf, overexpression of the *fli1a*-NICD:GFP construct did not significantly increase *runx1* in wild type embryos, but could rescue its expression in the *gata2a* ^{Δ i4/ Δ i4} mutant HE to a reasonable extent (new Supplementary Fig 5F). *Gata2b* expression was increased beyond wild type levels (but to a similar extent) in both wild type and *gata2a* ^{Δ i4/ Δ i4} mutants by the *fli1a*-NICD:GFP construct (new Supplementary Fig 5G). Taken together, these experiments establish that Gata2a is required for the initial expression of *runx1* in HE and that both *runx1* and *gata2b* are also regulated by Notch signalling acting independently of Gata2a.

The NICD overexpression and the DAPM treatment data suggest that the Notch input is sufficient to rescue *runx1* and *gata2b* expression in *gata2a* ^{Δ i4/ Δ i4} mutants. The Results and the Discussion sections have been modified to incorporate the new data.

- The authors do several experiments with the *gata2a*-i4 enhancer and they have identified a minimal sequence of 150bp that can actual drive *gata2a* expression in

endothelial cells. Can the authors perform some mutations on transcription factor motifs to identify the factors that regulate the activity of this enhancer?

We agree with the reviewer that it is important to understand how *gata2a* expression is regulated through the *i4* enhancer. Some of this work has been done for the mouse +9.5Kb enhancer where it was shown that deletion of 3 ETS sites decreased activity of the enhancer, while deletion of the E box motif extinguished all enhancer-driven expression of a lacZ reporter gene (Khandekar et al, 2007, Development). In addition, it is thought that Gata2 itself binds to the E-box/GATA motif to positively regulate its own expression through the +9.5 enhancer in a positive feedback loop (Gao et al, 2013, J Exp Med). A search for known motifs using JASPAR software (not shown) has indicated the presence of a high number of putative transcription factor binding sites that could contribute to the activity of the enhancer. Although we agree that this merits further investigation, we feel is outside of the scope of this manuscript.

- The authors detected AML in one of their three mutants. This is very interesting. Could they use more adult fish and observe what is the propensity of this? It would be great if the authors could prove that these blasts that they observe are transplantable.

We have now analysed the haematopoietic cell morphology of an additional 7 *gata2a* ^{$\Delta i4/\Delta i4$} mutants (total=10) and found no further evidence of increased excess blasts in the WKM, suggesting that the propensity to develop this phenotype is 10% (1 in 10) or less. As detailed above as a reply to reviewer #1, transplantability is an important criterion to establish whether the blast-like cells we see in the WKM of *gata2a* ^{$\Delta i4/\Delta i4$} mutants are AML. The estimated low penetrance of these 'AML-like' phenotype precludes us from performing transplantation experiments reliably to assess whether indeed the excess blasts are AML or not and we have modified the text to take these limitations into account.

Minor points

- In the abstract (p.2 line 24) the authors claim that their mutants developed acute myeloid leukemia. This is a very strong statement that should be softened to comply with the actual data.

We thank the reviewer for pointing this out. This is indeed a strong statement and is a concern that was raised by reviewers #1 and #2. To address these concerns, we have now modified our text to reflect the actual data (please see the response to point 4 raised by reviewer #1).

- P6 line 124. The statement in this phrase is wrong. ATAC-seq provides regions of open and not necessarily active chromatin.

We apologise for the phrasing that we recognise as ambiguous. We have now modified the sentence to help clarify that ATAC provides only open chromatin regions.

- Why did the authors perform the motif analysis only on enhancer regions? It is well established that GATA factors bind primarily to enhancers. Can the authors expand their motif search?

The motif analysis was performed exclusively as a 'quality control' for the ATACseq experiment to ensure that we would observe the expected enrichment for ETS binding sites in endothelial cells. We have now expanded the motif search to include all of the differentially 'expressed' peaks and indeed found the expected enrichment in ETS binding sites in endothelial cells. To illustrate this, the new data is now presented as two short lists ranking the top 10 motifs found enriched in endothelial cells and the top 10 motifs found in peaks common to endothelial and non-endothelial cells. These two lists replaced the previous panel I in Supplementary Fig. 1.

- The authors claim that four peaks were differentially accessible in the *gata2a* locus. Can the authors provide any information on the other peaks? Why did they choose the i4 enhancer?

The other peaks enriched in ECs in the *gata2a* locus are in intron 2, exon 4 and an intergenic peak 20kb downstream (in addition to the promoter peak). We chose the i4 enhancer because it was the only putative enhancer region in the *gata2a* locus that showed sequence conservation compared to human and mice (see Fig 1B), and an open chromatin peak. In addition, the region corresponding to the i4 enhancer in mice and humans (+9.5 enhancer) is known to regulate endothelial expression of *gata2*, suggesting that its deletion in zebrafish would lead to loss of *gata2a* in endothelium – exactly what we wished to achieve to study Gata2a in HE programming. None of the other peaks in the locus satisfied these criteria.

- The authors show that their mutants have reduced *gata2b* expression. Can this result be recapitulated by simply knocking down *gata2a*?

Gata2a is expressed early on in ectoderm and later in neural tissue as well as in the endothelium and erythroid precursors. The *gata2a*^{Δi4/Δi4} mutant is an excellent example of how deletion of an enhancer element can regulate gene expression in a tissue specific manner. We feel that knocking down *gata2a* (with a MO, for example) would not be an adequate substitute and would not add substantial value to our manuscript. Interestingly, a recent paper has shown that a different deletion of the same enhancer leads to loss of endothelial *gata2a* (Shin et al, 2019, Dev Cell), serving as independent validation of our approach.

- The authors should provide some more statistics. For example, on figure 5 they use a sub-optimal amount of *gata2b* morpholino, but in their quantification they do not present the statistics between wild type and wild type injected with the morpholino. Similarly and on the same figure, they show that ectopic expression of *gata2b* increases *runx1* expression in *gata2αΔi4/Δi4* mutants. They also claim this rescue reaches the levels of *runx1* expression in wild type fish, but this statistic comparison is missing from the figure. Is really the difference non-significant? Finally, on Figure S5, they claim that the absence of *gata2b* or both *gata2α* and *gata2b* has the same impact on *cmyb* expression at 33 hpf. Could they present the statistic comparison between these two and show it is not significant?

We have now modified the panel on fig 5B and D to show that the expression differences pointed out by the reviewer are indeed not statistically significant. We have also added the actual p values for those comparisons to the figure legends in the results section. To compare *cmyb* expression in *gata2b^{-/-}* against *gata2α^{Δi4/Δi4}*, *gata2b^{-/-}* mutants we re-analysed the data (one way ANOVA). This comparison yielded a p value of 0.93 (Dunnnett's T3 test for multiple comparisons, $t=1.616$, d.f.=9.817, $p=0.93$), thus confirming that *cmyb* expression levels in *gata2b^{-/-}* mutants at 33hpf is not significantly different from that of *gata2α^{Δi4/Δi4}*, *gata2b^{-/-}* mutants. The extra information was added to the Supplementary Figure (now Supplementary Fig 4, panel F, new comparison labelled n.s.); the actual details of the comparison as detailed above were added to the figure legend.

- Although they are quite meticulous describing their experiments and the numbers of embryos used in their figure legends, the authors should also include numbers in all their zebrafish experiments (e.g. in situ hybridization experiments). For instance, these numbers are missing on figure 5A,C,F and supplementary figure 5B-D. We thank the reviewer for pointing this out; we have added the number of embryos analysed to each of the panels in Figures 5A and F and to the graph on Supplementary Figure 5A (now Supplementary Fig. 4F); panels 5B-D (now Supplementary Fig. 4G-I) are meant as an example of one of the embryos of the genotypes shown and quantified in panel A. The images in Figure 5C aim to illustrate how we scored the 'wild-type-like' versus 'reduced' *cmyb* staining in this experiment. Thus, the numbers corresponding to each experimental condition are shown inside each bar of the graph in panel 5C. To clarify this further, we now added the total number of embryos analysed per condition to the bottom of the graph (below each bar) on Fig. 5, panel 5C.

- Although no doubt is cast on the validity of the results, the number of embryos the authors present on pictures of in situ hybridization is relatively low. Could they examine a bit more embryos (at least regarding their basic phenotypes in *gata2αΔi4/Δi4* mutants) and include them on the figures?

We have taken the reviewer's comments on board and examined more embryos for *gata2a* and *kdrl* expression (numbers updated in Fig. 2A-D). We have also analysed more embryos and amended the numbers in Figure 3 panels A-B and E-F to clarify the total number of embryos analysed per genotype. In addition, we analysed *runx1* expression in the *gata2a*^{Δi4/Δi4} mutants at 24hpf (new Supplementary Figure 4A-C) to show that the phenotypes are consistent and reinforce that *runx1* expression in HE is affected already at its onset in these mutants. Supplementary Figure 4D and 4E show extra experiments quantifying *gata2b* in wildtype and *gata2a*^{Δi4/Δi4} mutants at 26hpf and 30hpf. These reinforce the existing data on Figure 3 and clearly demonstrate the recovery of *gata2b* expression in the *gata2a*^{Δi4/Δi4} mutants by 30hpf. In Figure 4, we have now added the number of embryos analysed for *runx1* expression in wildtype and mutant at 48hpf in panel B for clarification (see new Figure 4). We have also analysed more wildtype and mutant embryos stained for *rag1* at 4 and 5dpf and updated Figure 4E-H accordingly.

- Could the authors check whether (except for *kdrl*) the expression of the arterial marker ephrinβ2 changes in *gata2a*^{Δi4/Δi4} mutants?

We thank the reviewer for their insight as it has been shown that EfnB2 regulates haematopoiesis from the dorsal aorta (Chen et al, 2016, Scientific Reports). We have now performed a qPCR for *efnb2a* at 23 and 30hpf and found that its expression is reduced in endothelial cells at 23hpf but recovers by 30hpf. This suggests that Gata2a is required for its regulation, at least partially. Since *efnb2a* is regulated by Notch signalling (Lawson et al, 2001, Development), its expression recovery likely reflects the Notch-mediated recovery of haematopoiesis that we find in the *gata2a* mutants. This new data was added to the results section and to the new Supplementary Figure 3 as panel G.

- On figure 4G,H the embryos are, according to the manuscript, 5dpf embryos, but instead they are labelled as 4dpf.

Thank you for bringing this to our attention; we have now corrected the mis-labelled panels G and H to read 5dpf instead of 4dpf.

Reviewer #3 (Remarks to the Author):

This manuscript shows the exquisite regulation of hematopoietic development by the 2 Gata2 genes (2a and 2b) in Zebrafish embryos. The authors describe the conserved enhancer in the 4th intron of Gata2a and carefully manipulate it through Crispr/Cas9 methodology. These embryos are examined at several (mainly two) time points and effects on endothelial cell, hemogenic endothelial and hematopoietic cell development are investigated. Because the Kdrl GFP reporter is used, endothelial

cells can be sorted and expression analyses carried out. Moreover, combinations of transcriptome and protein expression analyses on zebrafish embryos validates and strengthens the data presented. The authors clearly show that the two Gata2s act at different stages of development and that Gata2b is regulated by two different inputs - by Gata2a and Notch. Finally, they suggest that Gata2a is required in adult hematopoiesis and that its deficiency presents a phenotype very much like Gata2 haploinsufficiency in the human MonoMac syndrome. Altogether the studies are well-performed, statistical analysis is appropriate and provide novel findings that advance the field. It presents important new insights into the complexity of Gata2 expression and function that previously have been difficult to reconcile in studies of mouse and human Gata2.

We thank the reviewer and are pleased to note they find that we provide novel findings into the complexity of Gata2 function *in vivo*.

Minor comment

Some data figure legends do not contain information on number of cells examined. We have reviewed the figure legends, also adding extra information required by reviewers #1 and #2 so we hope these issues are now corrected.

REVIEWERS' COMMENTS:

Reviewer #1 (Remarks to the Author):

The revisions have significantly strengthened the interesting manuscript. However, one component of new text is inaccurate and requires elimination or significant modification. The authors indicate on page 15:

In both cases the experiments were performed at E11.5, well after the HE357 had been established and given rise to HSCs 5. At E11.5 the numbers of haematopoietic cells in the AGM 358 were roughly comparable to their wild type siblings 19,26, raising the possibility that an initial HE defect

16 might have gone unnoticed in those studies. Thus, by analysing th 359 e contribution of two zebrafish Gata2

360 paralogues, we uncovered a previously unappreciated contribution by a Gata2 gene in the programming of

361 HE prior to HSC specification."

This statement requires correction. Gao et al. JEM used direct imaging approach to analyze emergence of HSC-containing cluster in the AGM at E10.5. The enhancer deletion abrogated emergence, based on quantitative analysis. The significance of these imagine results were confirmed by long-term transplants. Thus, the statement above is inaccurate and should be deleted. The result is not "previously unappreciated" but is consistent with expectations from previously published work - but importantly, extends the findings to a genetic model system.

Reviewer #2 (Remarks to the Author):

The authors have performed a substantial number of new experiments and have streamlined the manuscript according to the reviewers' comments. In my opinion the manuscript is greatly improved and the authors have answered my comments.

I would like to add that reduction in the size of the introduction and discussion will make the manuscript more succinct. Also on page 3 line 54 it is stated that HSCs are born in the AGM at 34hpf. A more accurate statement would be that HSCs are born in the AGM between 28 and 48hpf.

Response to the Reviewers' comments

We are pleased that both reviewers found the manuscript was greatly improved. Please see our detailed responses below.

Reviewer #1 (Remarks to the Author):

The revisions have significantly strengthened the interesting manuscript. However, one component of new text is inaccurate and requires elimination or significant modification. The authors indicate on page 15:

In both cases the experiments were performed at E11.5, well after the HE357 had been established and given rise to HSCs 5. At E11.5 the numbers of haematopoietic cells in the AGM 358 were roughly comparable to their wild type siblings 19,26, raising the possibility that an initial HE defect 16 might have gone unnoticed in those studies. Thus, by analysing th 359 e contribution of two zebrafish Gata2 360 paralogues, we uncovered a previously unappreciated contribution by a Gata2 gene in the programming of 361 HE prior to HSC specification."

This statement requires correction. Gao et al. JEM used direct imaging approach to analyze emergence of HSC-containing cluster in the AGM at E10.5. The enhancer deletion abrogated emergence, based on quantitative analysis. The significance of these imagine results were confirmed by long-term transplants. Thus, the statement above is inaccurate and should be deleted. The result is not "previously unappreciated" but is consistent with expectations from previously published work - but importantly, extends the findings to a genetic model system.

In light of the reviewer's comment on the accuracy of the text in the discussion, and the suggestion by reviewer #2 to shorten both the introduction and discussion sections, we have eliminated this paragraph altogether.

Reviewer #2 (Remarks to the Author):

The authors have performed a substantial number of new experiments and have streamlined the manuscript according to the reviewers' comments. In my opinion the manuscript is greatly improved and the authors have answered my comments.

I would like to add that reduction in the size of the introduction and discussion will make the manuscript more succinct. Also on page 3 line 54 it is stated that HSCs are born in the AGM at 34hpf. A more accurate statement would be that HSCs are born in the AGM between 28 and 48hpf.

We have now deleted some of the introduction and discussion to address the reviewer's comment and also modified the statement about the birth of HSCs in the AGM.